# Haploinsufficient phenotypes promote selection of PTEN and ARID1A-deficient clones in human colon

Nefeli Skoufou-Papoutsaki [1,2], Sam Adler[1], Shenay Mehmed[1], Claire Tume[1], Cora Olpe[1],
Edward Morrissey[3], Richard Kemp [1], Anne-Claire Girard[1], Elisa B Moutin[1],
Chandra Sekhar Reddy Chilamakuri[1], Jodi L Miller [1], Cecilia Lindskog [4], Fabian Werle[1], Kate Marks[5],
Francesca Perrone[2], Matthias Zilbauer[2], David S Tourigny[6] & Douglas J Winton [1,2]✉

## Abstract

**Cancer driver mutations are defined by their high prevalence in cancers and presumed rarity in normal tissues. However, recent studies show that positive selection in normal epithelia can increase the prevalence of some cancer drivers. To determine their true cancer-driving potential, it is essential to evaluate how frequent these mutations are in normal tissues and what are their phenotypes. Here, we explore the bioavailability of somatic variants by quantifying age-related mutational burdens in normal human colonic epithelium using immunodetection in FFPE samples ($N = 181$ patients). Positive selection of variants of tumour suppressor genes *PTEN* and *ARID1A* associates with monoallelic gene loss as confirmed by CRISPR/Cas9 mutagenesis and changes in their downstream effectors. Comparison of the mutational burden in normal tissue and colorectal cancers allows quantification of cancer driver potency based on relative representation. Additionally, immune exclusion, a cancer hallmark feature, is observed within ARID1A-deficient clones in histologically normal tissue. The behaviour resulting from haploinsufficiency of *PTEN* and *ARID1A* demonstrates how somatic mosaicism of tumour suppressors arises and can predispose to cancer initiation.**

**Keywords** Haploinsufficency; Clone Dynamics; Normal Tissue; PTEN;
ARID1A
**Subject Categories** Cancer; Genetics, Gene Therapy & Genetic Disease;
Signal Transduction

## Introduction

The definition of a cancer driver gene largely arises from the high prevalence of its mutated forms in cancer compared to their presumptive low frequency in normal epithelium. However, recent studies have shown that the prevalence of somatic mutations in normal epithelia can be highly skewed due to positive selection resulting in substantial clonal expansions. Importantly, selection acting to elevate mutational burden in normal tissue is a poor predictor of cancer driver activity, with gene mutations being strongly selected in host tissue but almost absent in cancers, such as *NOTCH1* in the oesophagus (Abby et al, 2023; Martincorena et al, 2018), and vice versa. To better relate mutation prevalence in cancers to a role in cancer causation, we need to understand the extent to which somatic variants are bioavailable at the time of transformation, and could be instead acting as passengers in the process. Currently, there is a lack of conceptual approaches to enable such analyses, largely because of a lack of data about variant selection in normal tissues.

We have previously been able to quantify the processes that generate age-related mutational burden in normal human FFPE colonic epithelium by visualising somatic variants of a small number of X-linked genes in situ. Here, we sought to extend the approach to include autosomal genes that were tumour suppressors for colorectal cancer (CRC). Notably, positive selection biases were found to be associated with *PTEN* and *ARID1A*-deficiency, increasing their overall mutational burden. Immunodetection and CRISPR/Cas9 mutagenesis related such phenotypes to the loss of a single allele, suggesting a haploinsufficient role for these tumour suppressors in a pre-neoplastic context. Knowledge of selection biases and somatic variant burden in the normal colon allows a quantitative contribution of of passenger effects for cancer drivers based on their representation in normal tissue.

Relatedly, the spatial localisation of somatic variants allows the phenotyping of clones in a way that has not been attempted in normal human tissue previously. This permits an assessment of whether known cancer or pre-cancer lesion hallmarks (Hanahan and Weinberg, 2011; López-Otín et al, 2023; Stangis et al, 2024), can already be observed in mutant clones in normal tissue. Their presence might not only be indicative of the nature of selection but also serve to discriminate cancer drivers from non-cancer driver events. Applying such an approach to ARID1A mutants identified

[1]CRUK Cambridge Institute, University of Cambridge, Cambridge CB2 0RE, UK. [2]Wellcome-MRC Cambridge Stem Cell Institute, University of Cambridge, Cambridge CB2 0AW, UK. [3]MRC Weatherall Institute of Molecular Medicine, University of Oxford, John Radcliffe Hospital, Headington, Oxford OX3 9DS, UK. [4]Department of Immunology, Genetics and Pathology, Cancer Precision Medicine Research Program, Uppsala University, Uppsala 751 85, Sweden. [5]University of Leeds School of Medicine, Leeds Institute of Medical Research, Pathology and Data Analytics, University of Leeds, Leeds LS2 9JT, UK. [6]School of Mathematics, University of Birmingham, Edgbaston B15 2TT, UK. ✉E-mail: Doug.Winton@cruk.cam.ac.uk

local exclusion of immune cells, a recognised cancer hallmark, as a feature of histologically normal epithelial clones.

# Results

## Meta-analysis of truncal CRC drivers identifies genes involved in cancer initiation

Bulk sequencing of cancers has identified many genes with driver mutations for CRC, occurring at any stage of cancer progression. To focus on genes involved in cancer initiation, we collated known truncal clonal (as opposed to subclonal) drivers identified via multi-regional sampling of individual cancers by performing a meta-analysis of available studies (N = 115 tumours) (Sottoriva et al, 2015; Househam et al, 2022; Uchi et al, 2016; Saito et al, 2018; Roerink et al, 2018; Banerjee et al, 2021). Truncal CRC driver mutations were identified by the dN/dS ratio (dN/dS >1) for CRC (Martincorena et al, 2017) (Fig. 1A; Appendix Fig. S1a–d). Most tumours (70%) had 2–4 truncal hits in recognised cancer driver genes, irrespective of their *APC* status (Fig. 1B,C). While most of the single-hit tumours were *APC*-driven, this accounted for only 3% of all tumours (Fig. 1D,E). Importantly, among tumours with at

least three driver mutations, a similar number of tumours with alternative combinations of drivers to the classical CRC combination (*APC*, *KRAS*, *TP53*) (Fearon and Vogelstein, 1990) were observed (45 and 39%, respectively) (Fig. 1F). Events previously classified as late or subclonal, such as *SMAD4* or *PTEN* (Gerstung et al, 2020), were also occasionally found in the trunk of phylogenetic trees, either co-occurring with *APC* or *KRAS*, or not. Mutated genes identified as iteratively truncal events (>2 occurrences) with truncating mutations, were selected as candidates for antibody screening aimed at immunodetection (Fig. 1G).

## Loss of staining in tumour suppressor genes *PTEN*, *SMAD4* and *ARID1A* detected in the human colon

Previously, we used in situ immunodetection of X-linked genes in normal FFPE colonic tissue to detect clones that had lost the active allele and infer neutral or biased clonal dynamics (Nicholson et al, 2018; Olpe et al, 2021). Extending this approach to genes from the meta-analysis, an antibody screen was performed for nine tumour suppressor proteins encoded by genes commonly inactivated by truncating mutations in CRC (Fig. 1H). Detecting loss of function as a clonal event requires strong, consistent immunoreactivity across the crypt axis. Many targets failed if immunoreactivity

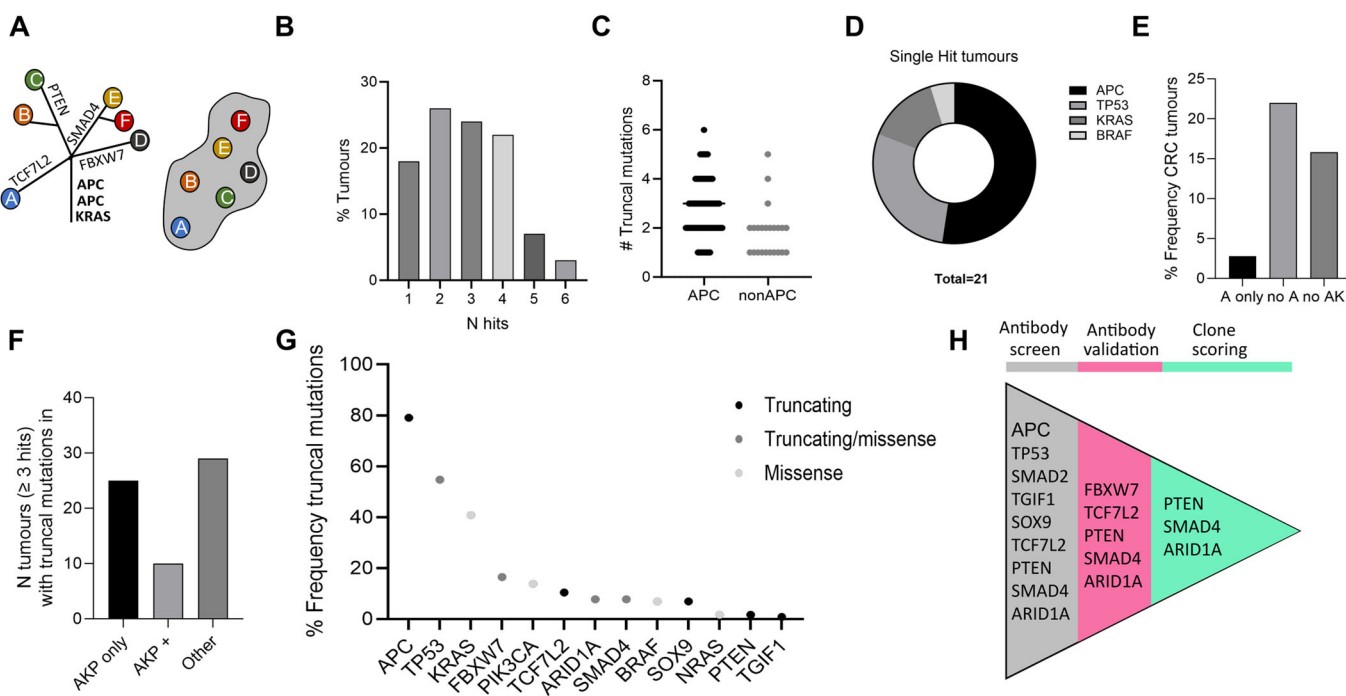

**Figure 1. Meta-analysis of multi-regional sampling studies identifies truncal mutations in CRC that formed the basis of an antibody screen.**

N = 115 tumours were analysed from five independent studies. (Banerjee et al, 2021; Uchi et al, 2016; Roerink et al, 2018; Househam et al, 2022; Sottoriva et al, 2015.) Considers CRC driver genes as defined by dN/dS >1 (Martincorena et al, 2017). (A) Example of truncal analysis schematic. Grey mass represents tumour which has been sampled in six regions A–F. Mutations found in all parts of the tumour sampled are placed on the trunk of the tree representing the last common ancestor. Other mutations define subclones within the tumour. (B) Distribution of truncal hits. (C) Number of truncal mutations in tumours with or without APC mutations. (D) Composition of single-hit tumours. (E) Percentage frequency of tumours with truncal mutations. A: APC, AK: APC KRAS. (F) For tumours with three or more hits, the genetic combinations found in the trunk are shown. AKP only: only hits in APC, KRAS and TP53, AKP+ : APC, KRAS and TP53 plus another driver, Other: any other combination of events. (G) Percentage frequency of tumours with truncal mutations. For clonal tracking by immunohistochemistry only truncating mutations can be used. (H) Pipeline for identification of cancer driver genes for which loss of staining can be detected using immunohistochemistry and can be used to track clones. List in grey shows targets selected for antibody screen. Pink indicates targets that failed at the antibody validation step. Antibodies for PTEN, ARID1A and SMAD4 in green passed all validation steps and around 100 patients per mark were screened for deficient clones. Source data are available online for this figure.

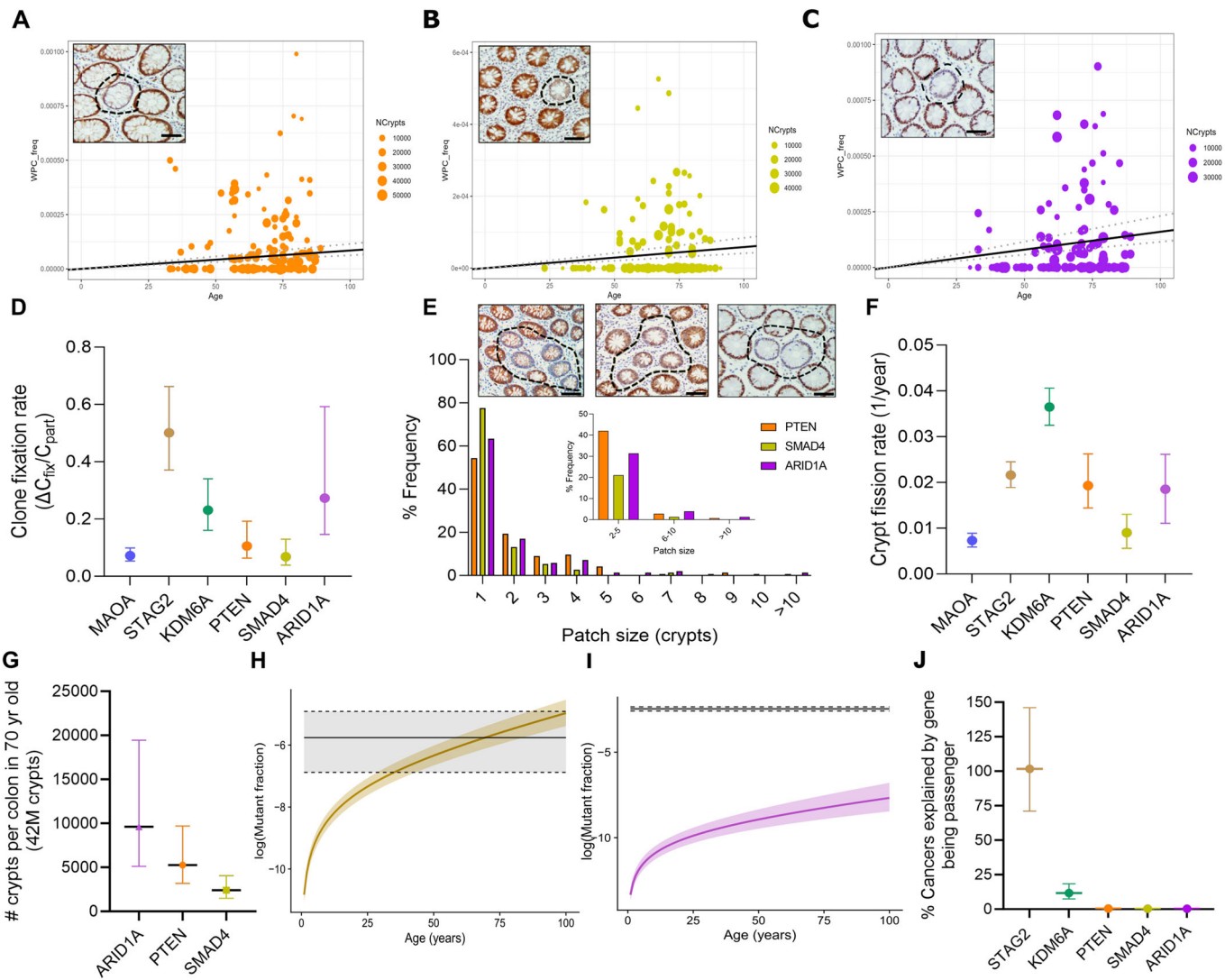

**Figure 2. Inference of clonal dynamics for tumour suppressor genes PTEN, ARID1A and SMAD4.**

(A) PTEN wholly populated crypt (WPC) frequency plotted against patient age with representative example. Inset image indicates a PTEN-deficient clone. (B) SMAD4 WPC frequency plotted against patient age with representative example. Inset image indicates SMAD4-deficient clone. (C) ARID1A WPC frequency plotted against patient age with representative example. Inset image indicates an ARID1A-deficient clone. (D) Comparison of intra-crypt dynamics/clone fixation rate inferred by RHClones package for X-linked genes MAOA, STAG2 and KMD6A and autosomal genes PTEN, SMAD4 and ARID1A. Calculated as the ratio of the slope of WPCs ($\Delta$Cfix)/ frequency of partially populated crypts (Cpart). (E) Patch size distribution and representative examples of PTEN, SMAD4 and ARID1A patches in order. Inset images indicate PTEN, SMAD4 or ARID1A-deficient multi-crypt patches. (F) Comparison of crypt fission rates, per year inferred by RHClones package for X-linked genes MAOA, STAG2 and KMD6A and autosomal genes PTEN, SMAD4 and ARID1A. (G) Number of mutant crypts per colon in 70-year-old individual. (H, I) Examples of mutational burden in normal tissue versus frequency in cancer analysis. Predicted mutational burden in normal tissue based on clone dynamics (coloured curve) compared to the frequency of truncating mutations for that gene in colorectal cancer in COSMIC (black line). Shading indicates 95% credible intervals and dotted lines indicate 95% confidence intervals. (H) STAG2. (I) ARID1A. (J) The percentage of tumours explained by the gene being a passenger was calculated as the ratio of normal burden versus tumour burden at age 70. Data information: PTEN: $N = 103$ patients and 2,492,712 total crypts, SMAD4: $N = 88$ patients and 1,661,059 crypts, ARID1A: $N = N = 101$ patients, $N = 1,990,179$ crypts. Data presented as mean. All error bars, dotted black lines or shading indicate 95% credible intervals from mathematical inferences. All scale bars indicate 50 μm. Source data are available online for this figure.

precluded detecting clonal loss of signal (Fig. EV1A–C). Antibodies for five targets with homogenous staining were run against ten FFPE blocks from patients aged 60–80. Candidate clones with reduced immunoreactivity in contiguous crypts were validated by confirming loss of immunoreactivity in knockout mice and/or cells and with independent antibodies on serial sections (Fig. EV1D,E). The analysis identified epithelial clones with reduced PTEN, SMAD4 and ARID1A immunoreactivity (Fig. 2A–C).

## Assessment of biases in clone fixation

Stem cell clones become permanently fixed in the colon when they populate an entire crypt, a process known as monoclonal conversion (Cook et al, 2000; Winton and Ponder, 1990; Winton et al, 1988). This fixation often results from neutral competition between a few clonogenic stem cells, though positive biases increase the probability of monoclonal conversion. The frequencies and

states of PTEN, SMAD4 and ARID1A-deficient clones [found as wholly populated crypts (WPC), partially populated crypts (PPC) and multi-crypt patches] were assessed by immunohistochemistry (IHC) on FFPE *en face* sections from around 100 individuals per gene (aged 30–91) (Appendix Table S1).

Linear regression revealed an age-related increase in WPCs for all three marks, with a slope ($\Delta C_{fix}$) of: $8.45 \times 10^{-7}$ for PTEN (95% credible intervals (CI): $6.26 \times 10^{-7}$–$1.15 \times 10^{-6}$); $5.87 \times 10^{-7}$ for SMAD4 (95% CI: $4.14 \times 10^{-7}$–$8.42 \times 10^{-7}$), and $1.60 \times 10^{-6}$ for ARID1A (95% CI: $1.15 \times 10^{-6}$–$2.31 \times 10^{-6}$) age (Fig. 2A–C). There was a trend of higher slopes being observed on the right side of the colon for all marks, particularly ARID1A, but this was not significant (Fig. EV2A–C).

The ratio of $\Delta C_{fix}$ to PPC frequency ($C_{part}$) quantitatively describes the bias towards monoclonal conversion, enabling comparison of clonal events independent of mutation rates (Nicholson et al, 2018). SMAD4-deficient clones display no advantage in intra-gland competition, comparable to the neutral loss of MAOA. In contrast, ARID1A-deficient stem cells display an increased fixation bias, most comparable to those with loss of KDM6A (Fig. 2D) (Olpe et al, 2021). A slight bias towards clone fixation was observed for PTEN-deficient stem cells, but with credible intervals overlapping that for MAOA.

## Assessment of biases in clone expansion via crypt fission

Clonal expansions forming multi-crypt patches of mutant crypts can arise via crypt fission, with fission rates inferred from the shape of the resulting patch size distribution across many patients (Fig. 2E). The homoeostatic crypt fission rate has been estimated at about 0.7% per year (95% CI: 0.0057–0.0087) (Nicholson et al, 2018), very similar to the fission rate for SMAD4-deficient clones (mean: 0.009, CI: 0.0056–0.0013) (Fig. 2F). In contrast, there was an approximately threefold increase in crypt fission rates inferred from PTEN and ARID1A-deficient clones (PTEN- mean: 0.019, 95% CI: 0.014–0.026, ARID1A- mean: 0.019, 95% CI: 0.011–0028).

Clone fixation and expansion via crypt fission together determine the burden of a given mutation in a normal colon by a certain age. In a 70-year-old individual, from inferred values of $\Delta C_{fix}$ and crypt fission rates, we can estimate that the entire colon (42 million crypts) would have ~10,000 ARID1A-deficient clones (mean: 9618, CI: 5124-19446), ~5000 PTEN-deficient clones (mean: 5250, CI: 3179-9702), and ~2500 SMAD4-deficient clones (mean: 2406, CI: 1491-4044) (Fig. 2G).

## Tissue representation allows the definition of cancer driver potential

Knowing the mutational footprint in normal tissue allows comparison with the frequencies of the same events in cancer, as a way to reclassify passenger and driver events based on their relative representation in both. As a motivating example, the mutational burden of the cancer driver *TP53* in normal oesophagus falls far below that observed in oesophageal cancer, while the mutational burden of *NOTCH1* is much higher in normal tissue, with the latter representing an extreme example of cancer-protective behaviour (Martincorena et al, 2018; Abby et al, 2023).

Simulated mutational burdens in normal colon using clone fixation and crypt fission rates for each gene were compared with

COSMIC CRC data (Tate et al, 2019) (Figs. 2H,I and EV2D–F). *STAG2* and *KMD6A* mutations confer positive biases in normal tissue but are not among the top CRC mutated genes. For *STAG2*, the predicted mutational burden in normal tissue matched that in CRC within a lifetime (69 years; range 13–110 years) (Fig. 2H), indicating that the frequency of inactivating *STAG2* mutations in CRC can be almost entirely attributed to positive selection in normal tissue. Consequently, by age 70, 100% (CI: 71 to 146%) of inactivating *STAG2* mutations in tumours can be explained as passenger events. In sharp contrast, around only 0.3% of the *PTEN*, *SMAD4* or *ARID1A* mutations in tumours (mean: 0.25–0.39) can be described as passengers in this way (Fig. 2J), suggesting they are true cancer driver genes positively enriched in CRC beyond their level of selection in normal tissue. KDM6A lies somewhere in between, whereby around 10% of KDM6A mutations in tumours could be explained as passenger events, suggesting a weaker driver potency.

## Deficient autosomal clones are the result of monoallelic loss

To assess whether changes in clone dynamics are linked to biallelic or monoallelic gene loss, multiplex immunofluorescence (IF) was used to quantify the intensity of immunostaining for deficient clones for autosomal (PTEN and ARID1A) or X-linked (MAOA, STAG2 and KDM6A) genes compared to adjacent WT crypts (Fig. 3A,B; Appendix Fig. S2a,b). Clones deficient for X-linked genes showed an 80–90% reduction in immunoreactivity due to functional hemizygosity, while clones deficient for autosomal genes showed a 40–50% reduction, consistent with monoallelic loss.

It has been previously suggested that, while the transcriptome reflects copy number alterations, the proteome may revert to a diploid state (Stingele et al, 2012). To link PTEN and ARID1A-deficient clones with monoallelic loss, laser-capture microdissection and targeted amplicon sequencing were used on deficient patches. Samples carrying truncating mutations within the coding region of *PTEN* or *ARID1A* correlated with a 50% reduction in protein intensity (Fig. 3C).

To confirm the effect of monoallelic loss on protein expression, CRISPR-Cas9 was used to introduce heterozygous mutations in *PTEN* and *ARID1A* in human intestinal organoids using a modified ribonucleoprotein (RNP) approach (Skoufou-Papoutsaki et al, 2023). Dead Cas9 protein was included in the RNP complex to limit the efficiency of genomic edits and maximise the generation of heterozygous knockouts (KOs). At a population level, editing efficiencies were similar for *PTEN* and *ARID1A* (around 40% of alleles) for their respective optimised ratios (Fig. EV3A,B). Individual organoids were isolated, and expanded prior to sequencing to establish their knockout status (Figs. 3D–F and EV3C). Wildtype, heterozygous, and homozygous *PTEN* cultures were identified in a 2:6:8 ratio, but no homozygous *ARID1A* knockouts were derived, with the proportion being: 13:4:0. This suggests that *ARID1A* homozygous KO organoids are incapable of clonal growth, which has also been shown previously with mouse spheroids (Hiramatsu et al, 2019). Heterozygous clones contained one WT and one edited allele (comprising frameshifts of either: +1, −7, −4) (Fig. EV3D,E).

Protein expression analysis via capillary immunoassay (Wes, Biotechne) on lysates from edited organoids showed that PTEN

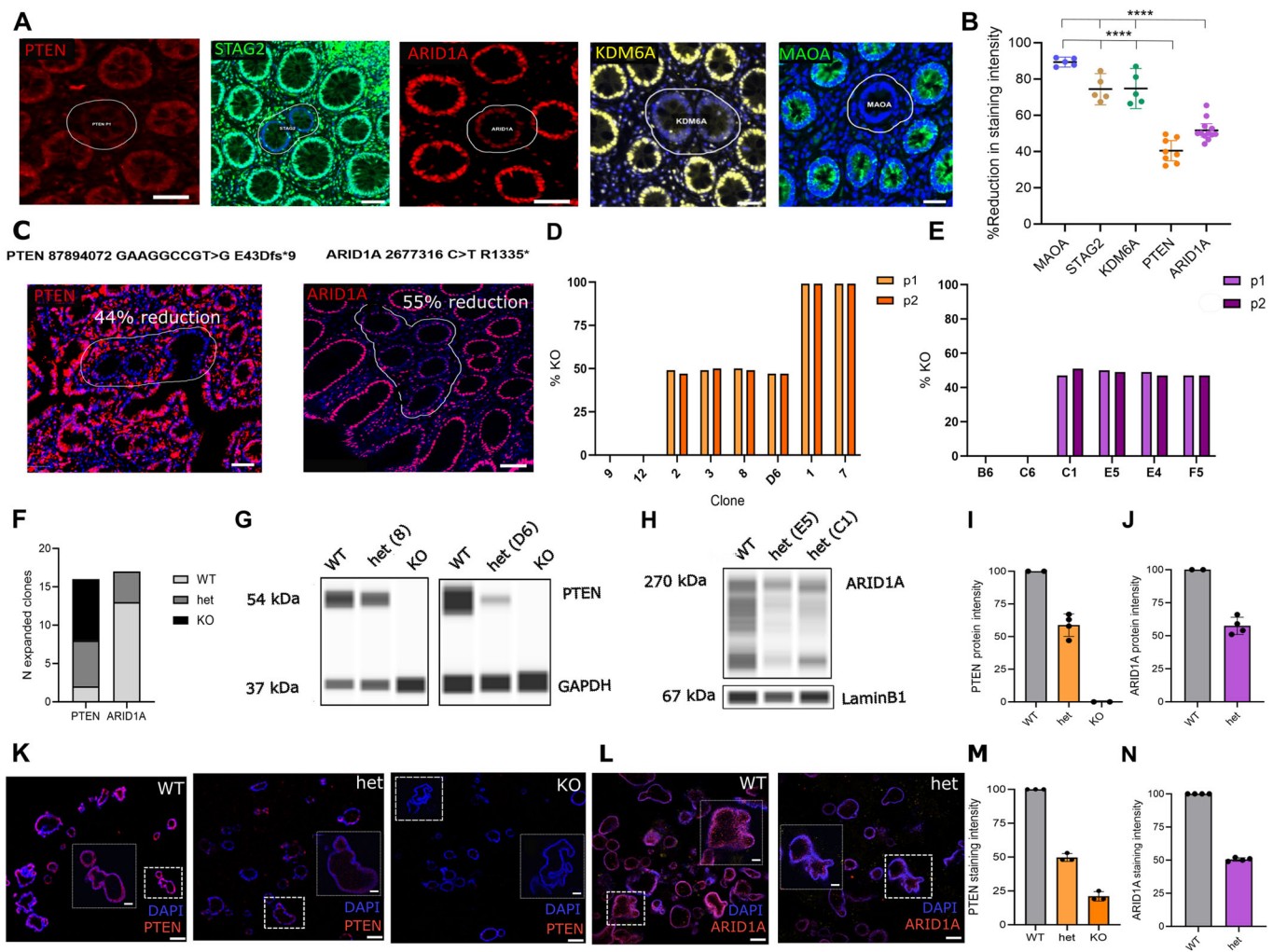

**Figure 3. Quantification of protein expression of autosomal-deficient clones indicates monoallelic inactivation.**

(A) Representative examples of detected clones with immunofluorescence. PTEN, STAG2, ARID1A, KDM6A, MAOA in order. (B) Quantification of staining intensity in X-linked gene clones (MAOA, KDM6A and STAG2) compared to autosomal gene clones (PTEN and ARID1A). $N = 5$ MAOA clones, $N = 5$ STAG2 clones, $N = 5$ KDM6A clones, $N = 8$ PTEN clones, $N = 12$ ARID1A clones. Corresponding colours represent statistical significance for PTEN or ARID1A comparisons. (C) Left- Example of PTEN-deficient patch with 44% reduction in staining intensity where heterozygous *PTEN* deletion (E43Dfs*9) was detected. Right- Example of ARID1A-deficient patch with 55% reduction in staining intensity where heterozygous *ARID1A* truncating mutation (R1335*) was detected. White line defines the mutant clone. (D) Percentage KO score of PTEN clones generated from single organoids. (E) Percentage KO score of ARID1A clones generated from single organoids. No full KO ARID1A clones were generated. (F) Genotypes of generated clones and their proportions. (G) Wes Biotechne capillary immunoassay for PTEN clones. (H) Wes Biotechne capillary immunoassay for ARID1A clones. (I) Quantification of G. $N = 2$ WT clones, $N = 4$ het clones, $N = 2$ KO clones. (J) Quantification of H. $N = 2$ WT clones, $N = 2$ het clones with 2 technical replicates. (K) Wholemount organoid images stained for PTEN. Genotype order: *PTEN* WT, *PTEN* het, *PTEN* KO. Inset: (white box) indicates expanded view of organoid (dashed box) (L) Wholemount organoid images stained for ARID1A. Genotype order: *ARID1A* WT, *ARID1A* het. Inset: (white box) indicates expanded view of oraganoid (dashed box) (M) Quantification of K. $N = 3$ WT, het and KO clones. (N) Quantification of L. $N = 2$ WT clones with two technical replicates, $N = 2$ het clones with 2 technical replicates. Data information: Data presented as mean ± SD. All scale bars indicate 50 μm. (B) Annova with multiple comparison testing was performed. ****$p < 0.0001$ for all comparisons of PTEN or ARID1A vs MAOA, STAG2 and KDM6A. Source data are available online for this figure.

protein was absent in homozygous KO organoids, while PTEN and ARID1A band intensities were reduced by 40–60% in heterozygous KO samples (Fig. 3G–J). Next, mimicking the in situ detection used in patient samples, organoids were fixed and stained as whole-mounts using immunofluorescence. This approach confirmed reduced staining (50%) in heterozygous *PTEN* and *ARID1A* KOs (Fig. 3K–N).

These findings demonstrate that monoallelic gene loss correlates with a 50% protein reduction for PTEN and ARID1A, confirming that these tumour suppressors can act in a haploinsufficient manner, with positive clone biases in the normal tissue arising from the loss of a single allele (phenotypic haploinsuffiency).

## Changes in downstream pathways in *PTEN* and *ARID1A* heterozygous organoids

The tumour suppressive function of PTEN is primarily mediated by negative regulation of the PI3K-Akt pathway. Previously, studying homozygous KO organoids, we showed an increase in p-AKT levels for *PTEN* mutant organoids (Skoufou-Papoutsaki et al, 2023).

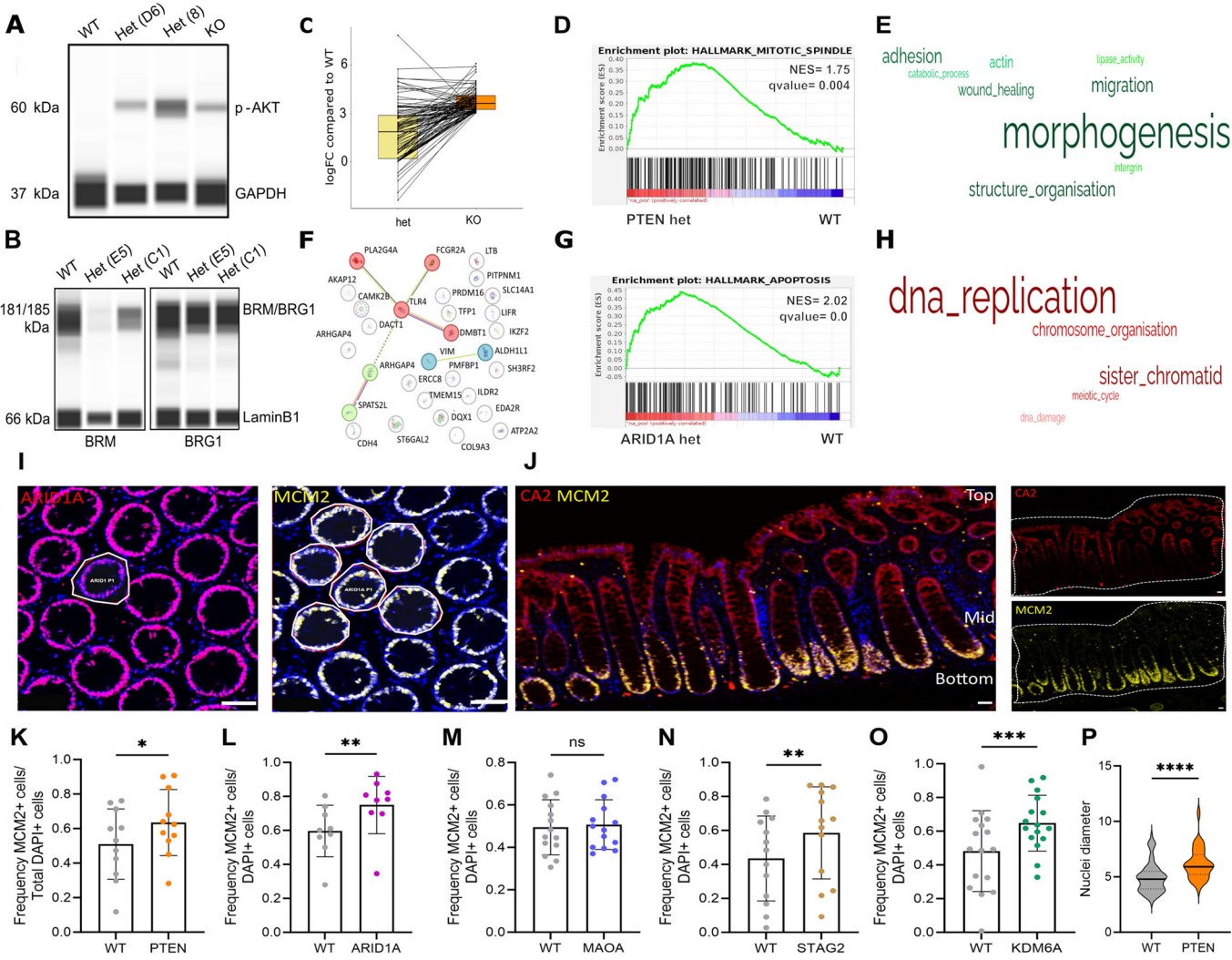

**Figure 4. Heterozygous mutations in PTEN and ARID1A organoids induce changes in downstream pathways and epithelial cell properties.**

(A) p-AKT Wes on *PTEN* edited organoids. (B) BRM and BRG1 Wes on *ARID1A* edited organoids. (C) Log fold change of het vs WT or KO vs WT for top 100 genes altered in PTEN KO organoids based on logFC. (D) Example of GSEA for relevant Hallmark pathway for positive enrichment in *PTEN* het vs WT comparison. (E) Word cloud of GOBP keywords from positively enriched gene sets in *PTEN* het vs WT. Created from Revigo scatterplot and simplification of GO term most accurately describing the cluster in a biologically meaningful way. The size of the word indicates how many gene sets were in that cluster. (F) Interaction network of significant differentially expressed genes in *ARID1A* het vs WT using STRING. (G) Example of GSEA for relevant Hallmark pathway for positive enrichment of *ARID1A* het vs WT. (H) Word cloud of GOBP keywords from negatively enriched gene sets in ARID1A het vs WT. Created from Revigo scatterplot and simplification of GO term most accurately describing the cluster in a biologically meaningful way. The size of the word indicates how many gene sets were in that cluster. (I) ARID1A and MCM2 staining on serial sections. Right image shows location of MCM2 counting on ARID1A-deficient clone and five WT neighbouring crypts. Dashed border indicates an ARID1A-deficient clone. (J) Markers used for the definition of crypt axis. CA2 is expressed at the crypt top and MCM2 is expressed at the crypt botrtom. CA2 + MCM2- crypt top, CA2 + MCM2+ crypt middle, CA2- MCM2+ crypt bottom. Dashed border indicates tissue outline. (K–O) Quantification of MCM2+ cells in deficient clones within the crypt bottom. Expressed as frequency out of total DAPI+ cells in the crypt. Dots indicate the number of clones analysed. N = 9–16 clones. (K) PTEN, (L) ARID1A, (M) MAOA, (N) STAG2, (O) KDM6A. (P) Nuclei diameter size (μm) quantified in WT or PTEN-deficient cells in FFPE tissue. n = 60–100 cells across nine clones. Data information: GSEA: genes ranked on fold change and standard error showing gene sets with FDR q value <1%. Data presented as mean ± SD. All scale bars indicate 50 μm. (K–O) Wilcoxon paired t-test. (K) p = 0.0114, (L) p = 0.0039, (M) p = 0.455, (N) p = 0.0018, (O) p = 0.0009. (P) Mann–Whitney test. p < 0.0001. *p < 0.05, **p < 0.01, ***p < 0.001. Source data are available online for this figure.

Capillary immunoassay confirmed that p-AKT was also elevated in *PTEN* heterozygous KOs (Fig. 4A). Similarly, the multiprotein SWI/SNF chromatin remodelling complex that contains ARID1A acts to regulate one of two catalytic subunits, BRM and BRG1, and only the former was seen to be reduced in *ARID1A* homozygous KOs (Skoufou-Papoutsaki et al, 2023). The same decrease only in BRM protein abundance was seen now in heterozygous *ARID1A* KO

organoids (Fig. 4B). This analysis confirms that monoallelic loss of *PTEN* and *ARID1A* is sufficient to perturb the principal downstream mediators of their function (molecular haploinsufficiency).

Next, bulk RNA-seq was performed to investigate if altered gene expression patterns linked to *PTEN* and *ARID1A* heterozygosity could identify processes mediating their selection advantages. For *PTEN*, only four genes were significantly differentially expressed

after multiple hypothesis testing corrections (*PLCXD2*, *LXOL4*, *TNNC1* and lncRNA transcript ENSG00000260401, $p < 0.05$). Despite the small number of differentially expressed genes (DEGs), transcripts in *PTEN* heterozygotes followed the pattern of homozygous KOs, with lower fold changes (Fig. 4C). Approximately 65% of the commonly detected genes (10,217 out of 16,090) exhibited the same directional changes. Gene set enrichment analysis (GSEA) on all detected genes ranked by fold change and standard error revealed several enriched pathways and biological processes (FDR <1%) (Appendix Fig. S3a–f). Hallmark pathway GSEA suggested that the competitive advantage of *PTEN* heterozygous cells might stem from increased proliferation, shown by upregulation of the mitotic spindle pathway and activation of fatty acid metabolism (Fig. 4D and Appendix Fig. S3a). Similar pathways were enriched with *PTEN* homozygous loss in organoids (Skoufou-Papoutsaki et al, 2023). In addition, a transcriptomic shift linked to developmental and regenerative processes, like organ morphogenesis, was observed (Fig. 4E). Of note, molecular haploinsufficiency has been previously shown in *PTEN* heterozygous endometrial organoids, with activation of PI3K and mTORC1 reported (Geurts et al, 2023).

Transcriptional profiling of heterozygous *ARID1A* organoids identified 39 DEGs ($p < 0.05$), four of which (*IF16*, *TLR4*, *FCGR2A* and *PLA2G4A*) showed strong downregulation (log2FC −2 to −7) and are part of a common immune-related cluster, as defined by STRING interaction analysis (Fig. 4F). GSEA for *heterozygous ARID1A* loss was unexpectedly linked to increased apoptosis and downregulation of DNA replication (FDR <1%) (Fig. 4G,H; Appendix Fig. S3e,f). Considered with the lack of viability of *ARID1A* homozygous KO organoids, this finding may suggest that survival and proliferation in heterozygous organoids are also adversely impacted.

These gene and protein expression changes show that heterozygous *PTEN* and *ARID1A* mutations can induce molecular haploinsufficiency in normal cells, potentially affecting their selection.

## Epithelial changes within deficient clones in situ

To relate the molecular and pathway adaptations seen in organoids to the positive biases observed in vivo, the epithelium of PTEN- and ARID1A-deficient clones was profiled for cellular changes using multiplex IF in FFPE sections (Fig. EV4A–Y). The proportion of cycling cells in the proliferative lower crypt region was determined based on expression of the DNA replication licensing factor MCM2 (Fig. 4I). The relative height along the crypt axis intersected by lateral sectioning was defined using MCM2 and CA2 expression, with the crypt base indicated as MCM2 + CA2- (Fig. 4J). Increased cell proliferation was observed in both PTEN- and ARID1A-deficient clones compared to WT neighbours, indicated by a higher frequency of MCM2 positive cells (Fig. 4K-O). The same phenotype was seen in clones with biased clonal behaviours, like STAG2 and KDM6A, but not in neutral MAOA-deficient clones. This analysis suggests that positive clonal biases are linked to increased cell proliferation. While proliferation seems to be required for an advantage, with reports of increased proliferation also being essential at the site of crypt bifurcation (Langlands et al, 2016), it might not be sufficient to explain it, with positional, adhesive and motility traits also

being known to play a role in clone dynamics (Ritsma et al, 2014; Azkanaz et al, 2022; Hageb et al, 2022).

Notably, ARID1A-deficient clones showed increased proliferation despite the adverse effects of *ARID1A* deletion in organoid culture. No increase in cell cycle arrest (pH3+) or apoptosis (cleaved PARP) was observed in ARID1A-deficient clones in situ (Fig. EV4G,V), indicating a lack of checkpoints triggered by ARID1A-deficiency. This suggests that context differentially influences epithelial phenotypes and highlights the need for selection biases to be determined in vivo rather than in model systems.

mTORC1 activation, a consequence of *PTEN* loss, can lead to larger cells (Fingar et al, 2002). To functionally assess PTEN-deficient clones in vivo, nucleus diameter was used as a proxy for cell size (Edens et al, 2013). PTEN-deficient cells exhibited larger nuclei compared to WT cells in primary FFPE tissue sections, suggesting the involvement of mTORC1 in their phenotype (Fig. 4P). The concordance between the seemingly advantageous phenotypes in organoids and the positive selection biases seen in clones in vivo, indicates a cell-intrinsic selective advantage from monoallelic PTEN loss.

## ARID1A-deficient patches create a microenvironment of immune exclusion

Motivated by the downregulation of immune response genes, and seeking to identify whole tissue features that might explain the differential impact of ARID1A-deficiency in vitro and in vivo, we phenotyped the stroma around ARID1A-deficient clones. A multiplex IF panel visualising mesenchymal cells, T-cells, B-cells, dendritic cells, neutrophils and macrophages was developed (Figs. 5A–C and EV5A–H; Appendix Fig. S4a–p). ARID1A-deficient patches exhibited reduced numbers of immune cells (CD45 + ) in both crypts and stroma (Fig. 5D,E). Specifically, significant reductions were seen in dendritic cells (CD11c+), B-cells (CD20+) and macrophages (CD68+) within the stroma of ARID1A-deficient patches compared to matched WT patches (Figs. 5F,G and EV5G). Both TIM3+ cell types, including T-cells (CD3+TIM3+), which mark IFNγ secreting or exhausted T-cells (Das et al, 2017), and macrophages (CD68+TIM3+), as well as CD163- macrophages (CD68+CD163−) were also reduced in ARID1A-deficient patches (Fig. 5H,I).

A recent elegant study showed that immune evasion in *ARID1A* mutant tumours is mediated by impaired chromatin accessibility of IFNγ responsive genes, including Th1 chemokines CXCL9, CXCL10 and CXCL11 (Li et al, 2020). To mimic this cytokine downregulation, blocking their cognate receptor, CXCR3, reduced immune cell infiltration and promoted tumour growth. Reviewing RNA-seq data from *ARID1A* heterozygous organoids revealed that CXCL11 and CXCL10 were the most highly downregulated chemokines, though this did not achieve significance after multiple hypothesis testing correction (Fig. 5J). However, this trend was confirmed via qPCR, with *ARID1A* heterozygous organoids showing nearly a twofold reduction in CXCL10 and CXCL11 RNA expression (Fig. 5K). Together, these findings demonstrate that ARID1A-deficiency leads to CXCL10 and CXCL11 downregulation in organoids, while Li et al showed this downregulation drives immune exclusion in colon cancer. Together this suggests

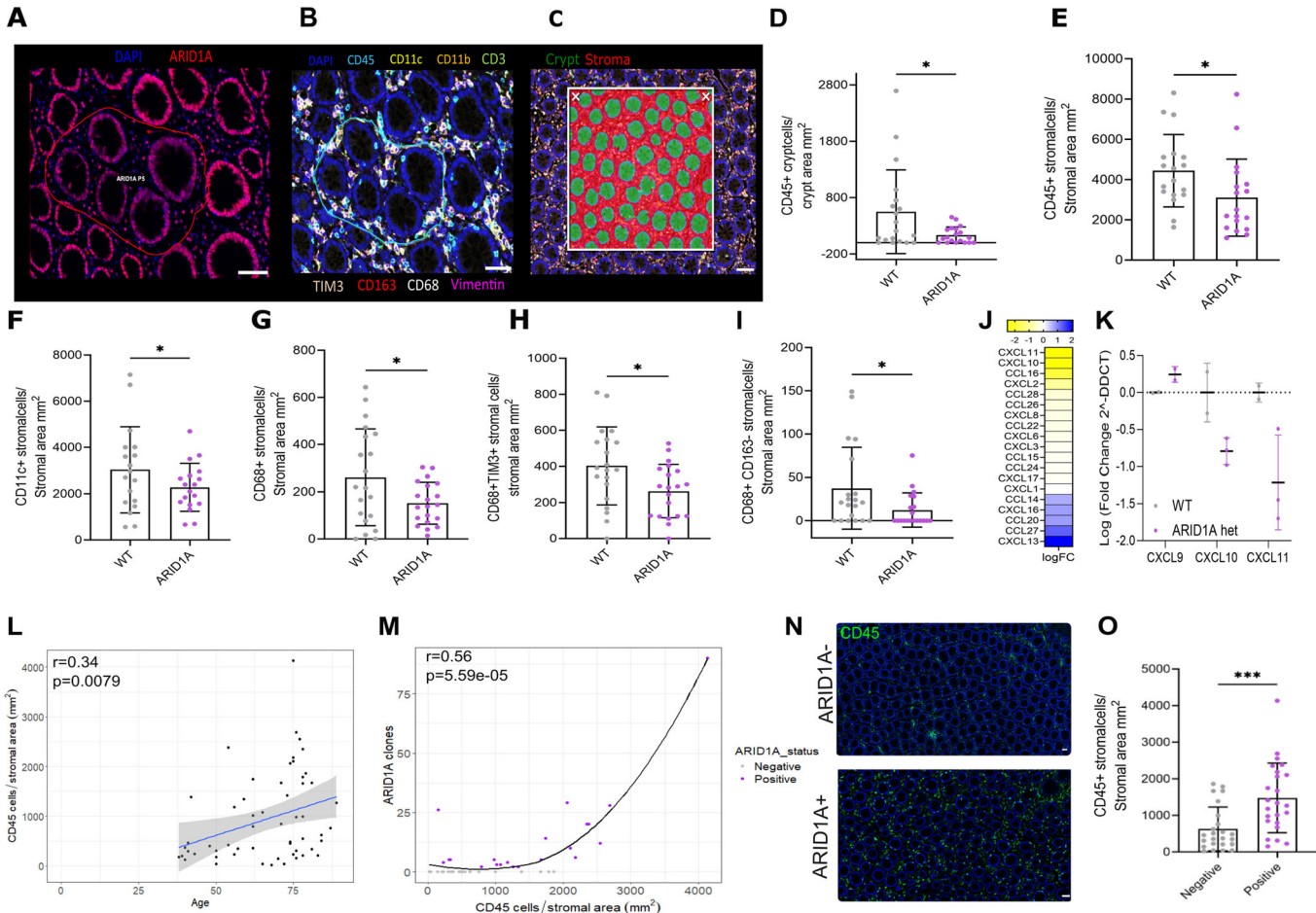

**Figure 5. ARID1A-deficient patches create an immune-exclusion microenvironment.**

(A) Illustrative example: locating a patch of five deficient ARID1A crypts for downstream characterisation. (B) Staining of deficient ARID1A patches for immune and mesenchymal cell markers. (C) Crypt-stroma random forest classifier trained using Halo (Indica Labs). (D) Quantification of CD45+ cells in the crypts using cell segmentation algorithm Halo (Indica Labs) of ARID1A patches (>5 crypts) and matched WT patch of the same area. Normalised against crypt area. (E–I) Quantification of positive cells in the stroma using cell segmentation algorithm Halo (Indica Labs) of ARID1A patches (>5 crypts) and matched WT patch of the same area. Normalised against stromal area. $N = 8$ patients, 23 clones. (E) CD45+, (F) Cd11c+, (G) CD68+, (H) CD68+TIM3+, (I) CD68 + CD163-. (J) Heatmap of logFC of ARID1A het vs WT organoids for all detected chemokines. (K) qPCR on IFNγ target chemokines on ARID1A het and WT organoids. (L) CD45+ cells per stromal area against patient age. $N = 57$ patients. (M) ARID1A clone count against CD45+ cells per stromal area. $N = 45$ patients. (N) CD45+ cells in tissue section of patients with (ARID1A+) or without (ARID1A−) clones. (O) Quantification of CD45+ stromal cells in patients with or without ARID1A clones. $N = 45$ patients. Data information: Data presented as mean ± SD. All scale bars indicate 50 μm. (D–I) Wilcoxon test. (D) $p = 0.026$ (E) $p = 0.012$, (F) $p = 0.034$, (G) $p = 0.044$, (H) $p = 0.032$, (I) $p = 0.049$. (L, M) Pearson correlation. (L) correlation = 0.34, $p = 0.0079$. (M) correlation = 0.56, $p = 5.585e-05$. (O) Mann–Whitney test. $p = 0.0006$. Source data are available online for this figure.

that the same axis could underly immune exclusion in normal colonic epithelium.

Immune dysregulation with age, leading to low-grade inflammation, has been widely reported (Franceschi and Campisi, 2014; Rasa et al, 2022; Franceschi et al, 2018; De Martinis et al, 2005). Hypothesising that a local immune exclusion environment could be preferentially selected in response to escalating immune infiltration of the tissue, we measured the proportion of CD45+ pan-immune cells infiltrating the stroma and confirmed their increase in frequency with age (Fig. 5L). Further, the number of ARID1A-deficient clones positively correlated with the number of CD45+ cells, and patients with ARID1A-deficient clones (ARID1A+) had a threefold higher number of pan-immune cells than age-matched patients without them (ARID1A-) (Fig. 5M–O).

Overall, these results suggest that epithelial ARID1A-deficiency acts non-cell autonomously to modulate the local immune environment and promote clonal expansion. Importantly, the cancer hallmark of immune modulation seen in ARID1A mutant tumours can already be observed in pre-neoplastic tissue, suggesting ARID1A-deficient fields may predispose to cancer.

# Discussion

Recognition of the extent of cancer driver mutations in the normal human colon has been achieved by sequencing approaches that are 'deep' in exploring whole exomes or genomes but involve relatively small amounts of tissue. This means that current insights are

largely qualitative, with only a few driver events directly detected. The likely presence of others is therefore extrapolated because of high de novo mutation rates, which are a feature of tissues with high cell turnover rates such as the colonic epithelium. Without substantial quantitative data describing mutation representation within normal tissues that is derived from; a large number of samples, large areas of tissue and at high sensitivity, it is impossible to establish if selection is acting upon some variants to increase their mutational footprint. Currently these approaches are only feasible if they are highly targeted to a defined and limited set of variants of interest. In responding to this challenge, we used an IHC approach targeting CRC tumour suppressor genes, enabling us to screen millions of crypts and detect mutations at a single crypt level.

An understanding of the mutational burden in normal tissue can allow the stratification of pro-oncogenic events into high or low-risk cancer drivers based on their representation in normal tissue and in cancer. For instance, clones with *PTEN* and *ARID1A* monoallelic loss are positively selected in normal tissue but far below the frequency observed in CRC, indicating additional selection in cancer. Conversely, *STAG2*, a less commonly mutated gene in CRC yet still previously classed as a cancer driver, seems to act as a passenger, with its frequency in cancer explained by its prevalence in normal tissue alone. While the haploinsufficiency of tumour suppressor genes is a well-described phenomenon, especially in the context of cancer development (Inoue and Fry, 2017), our study uniquely demonstrates how tumour suppressor genes can display molecular and phenotypic haploinsufficiency in normal human tissue. This has important implications regarding the search for other tumour suppressor genes that could be acting in similar ways to increase their bioavailability and potentiate transformation. Although we have only analysed a small number of targets, these capture both neutral and biased behaviours such that we were able to develop approaches that quantitatively expressed their driver (and passenger) status providing a route to evaluating cancer driver potency in the future.

A significant gap remains in understanding which cancer driver mutations found in CRC play a role in early cancer initiation. Bulk sequencing of cancers often includes events from all stages of cancer development. Multi-regional sampling studies have begun to address this issue but are limited in numbers. Our analysis of truncal mutations showed an equal number of tumours with frequently mutated drivers (*APC*, *KRAS*, *TP53*) and those with less frequent drivers, suggesting a level of genetic interchangeability between driver events in the early stages of cancer initiation. Additional data on early cancer development could help identify targets for evaluating positive selection in normal epithelium.

Beyond such probabilistic approaches of increased cancer risk due to increased mutational burden, clonal phenotyping may uncover if nascent forms of cancer hallmark dysregulation can be seen in clones arising from high-risk variants, prior to any overt pathology. Our in situ approach, revealed that ARID1A-deficient clones can create an environment of local immune exclusion that favours their selection in normal tissue. The correlation between the presence and number of ARID1A-deficient clones with the extent of immune infiltration may suggest that the degree of positive selection scales with the latter and with the severity of the inflammatory environment. The presence and selection of large ARID1A mutant clones identified in the context of inflammatory bowel disease support such an interpretation (Kakiuchi et al, 2020; Olafsson et al, 2020). Notably, the short-term advantage

provided by the immune exclusion phenotype associated with mutant *ARID1A* clones could also confer a pro-oncogenic function long-term (Li et al, 2020), explaining the role of *ARID1A* as a driver gene in colitis-associated cancer.

Overall, the present study presents a roadmap for future investigations aiming to understand the earliest events leading to the formation of cancer. With the advancement of spatial transcriptomics technologies, there will be a plethora of descriptive studies on tissues at steady state and in disease. Quantitative approaches, of the type employed here, will complement these studies, coupling the inference of cell fates and cancer driver potency with a mechanistic insight into the stratification of variants based on their risk of transformation.

# Methods

**Reagents and tools table**

| Reagent/resource | Reference or source | Identifier or catalogue number | Additional antibody info: Titre IHC/Wes |
|---|---|---|---|
| **Experimental models** | | | |
| Patient samples-FFPE tissue | Addenbrooke's Hospital Cambridge, Norwich University Hospital and St James University Hospital Leeds (REC 15/WA/0131, 06/Q0108/307 and 08/H0304/85 as well as 12/LO/1217, respectively) | | |
| Patient samples-primary organoid cultures | Addenbrooke's Hospital, REC 12/EE/0265. Organoids were modified to contain ARID1A and PTEN truncating mutations. | | |
| **Antibodies** | | | |
| ANLN Mouse | Atlas | AMAb90660 | 1:300 |
| ARID1A Rabbit | Cell Signalling | 12354 | 1:300/1:100 |
| ARID1A Rabbit | Atlas | HPA005456 | 1:200 |
| BRG1 Rabbit | Abcam | ab110641 | 1:2000 |
| BRM Rabbit | Cell Signalling | 6889 | 1:50 |
| CA2 Mouse | Santa Cruz | SC-48351 | 1:400 |
| CHGA Rabbit | Abcam | ab15160 | 1:200 |
| Cleaved-PARP Rabbit | Cell Signalling | 5625 | 1:400 |
| FBXW7 Rabbit | Abcam | ab109617 | 1:100 |
| FBXW7 Mouse | Atlas | CAB029987 | No suitable titre |
| FBXW7 Rabbit | Thermo Fisher | 40-1500 | 1:200 |
| GAPDH Rabbit | Cell Signalling | 5174 | 1:50 |
| KDM6A Rabbit | Cell Signalling | 33510 | 1:200 |
| LaminB1 Rabbit | Cell Signalling | 12586 | 1:2000 |

| Reagent/resource | Reference or source | Identifier or catalogue number | Additional antibody info: Titre IHC/Wes |
|---|---|---|---|
| MAOA Mouse | Santa Cruz | SC-271123 | 1:200 |
| MCM2 Mouse | BioRad | MCA1859 | 1:500 |
| MUC2 Rabbit | Santa Cruz | SC-15334 | 1:400 |
| p-Akt Rabbit | Cell Signalling | 4060 | 1:100 |
| pH3 Mouse | Cell Signalling | 9706 | 1:500 |
| PTEN Rabbit | Cell Signalling | 9559 | 1:300, 1:50 |
| PTEN Mouse | Sigma | 04-035 | 1:200 |
| SMAD4 Mouse | Santa Cruz | sc-7966 | 1:150 |
| SMAD4 Rabbit | Abcam | ab217267 | 1:400 |
| SOX9 Mouse | Atlas | CAB068240 | 1:800 |
| STAG2 Goat | LS-Bio | LS-B11284 | 1:1000 |
| TCF7L2 | Atlas | 34-3800/ CAB013535 | 1:200 |
| TCF7L2 Rabbit | Scientific Laboratory Supplies | SAB1404454 | 1:50 |
| Secondary HRP for TSA amplification Mouse | Thermo Fisher | T20916 | 1:200 |
| Secondary HRP for TSA amplification Rabbit | Thermo Fisher | 65-6120 | 1:200 |
| Secondary HRP for TSA amplification Goat | Thermo Fisher | A15999 | 1:200 |
| **Oligonucleotides and other sequence-based reagents** | | | |
| Fluidigm Primers | This Study | Appendix Table S3,4 | |
| **Software** | | | |
| ICE Synthego | https://ice.synthego.com/ | | |
| Halo Indica Labs | https://indicalab.com/halo/ | | |
| QuPath | https://qupath.github.io/ | | |
| RHclones package | https://github.com/ElEd2/RHClones | | |
| Amplicon Seq pipeline | https://github.com/crukci-bioinformatics/ampliconseq | | |
| **Other** | | | |
| TSA Reagent Cy5 | Akoya | TS-000203 | |
| TSA Reagent Cy3 | Akoya | TS-000204 | |
| TSA Reagent Alexa Fluorophore 488 | Thermo Fisher | B40953 | |
| Opal 6-colour manual detection kit | Akoya | NEL811001KT | |
| CD45 (Intracellular Domain) (D9M8I) and CO-0013-488 SignalStar™ Oligo-Antibody Pair | Cell Signalling Technologies | 81757 | |
| CD3ε (D7A6E™) and CO-0001-594 SignalStar™ Oligo-Antibody Pair | Cell Signalling Technologies | 84634 | |
| CD11c (D3V1E) and CO-0017-647 SignalStar™ Oligo-Antibody Pair | Cell Signalling Technologies | 96411 | |
| CD11b/ITGAM (D6X1N) and CO-0037-750 SignalStar™ Oligo-Antibody Pair | Cell Signalling Technologies | 67799 | |
| CD68 (D4B9C) and CO-0007-488 SignalStar™ Oligo-Antibody Pair | Cell Signalling Technologies | 73071 | |
| CD163 (D6U1J) and CO-0022-594 SignalStar™ Oligo-Antibody Pair | Cell Signalling Technologies | 43547 | |
| TIM3 (D5D5R™) and CO-0010-647 SignalStar™ Oligo-Antibody Pair | Cell Signalling Technologies | 15231 | |
| Vimentin (D21H3) and CO-0012-750 SignalStar™ Oligo-Antibody Pair | Cell Signalling Technologies | 61002 | |
| SignalStar™ Midplex IHC Buffer Kit | Cell Signalling Technologies | 29414 | |
| Arcturus PicoPure DNA extraction kit | Thermo Fisher | KIT0103 | |
| LP 8.8.6 integrated fluidic circuit (IFC) | Fluidigm | 101-7663 | |
| IVF EZ-Grip pipette | Cooper Surgical | 7-72-2800 and 7-72-2155/1, 7-72-2200/1, 7-72-2290/1 for tips | |

## Tissue processing and sectioning

Histologically normal colon tissue samples were obtained from cancer patients. Colectomy specimens were fixed in 10% neutral buffered formalin. From areas of tissue without macroscopically visible disease, mucosal sheets were removed from the specimens and embedded en face in paraffin (FFPE) blocks.

FFPE sections for staining were cut at 5 μm. To perform Laser-Capture Microdissection, sections were at 10-μm thickness onto special slides with PET membrane that had been irradiated with UV for 30 min prior to the cutting.

## Immunohistochemistry/immunofluorescence protocol

Immunohistochemistry was performed as previously described (Nicholson et al, 2018) (Reagents and Tools Table).

The immunohistochemistry protocol was adapted for immuno-fluorescence using secondary Alexa fluorophore conjugated anti-body (Thermo Fisher) and DAPI (1 μg/mL). Slides were scanned on the Akoya Biosciences Vectra Polaris Automated Quantitative Pathology Imaging System (Perkin Elmer).

## Multiplexing immunofluorescence

Here, multiplexing immunofluorescence refers to the immuno-fluorescence approach where multiple cycles of staining and antibody stripping are performed allowing staining of antibodies of the same species. Three different multiplexing protocols were used: the 3-plex standard tyramide signal amplification (TSA), the 6-plex TSA with Opal dyes (Akoya) 6-plex and the 8-plex Signal Star Technologies (CST). For the standard TSA, the dewaxing and antigen retrieval were performed as previously described (Nicholson et al, 2018). All incubation steps were performed at room temperature. The primary antibody was incubated for 30 min. The secondary antibody was Horseradish Peroxidase (HRP) conjugated and was incubated for 30 min. TSA reagent conjugated to standard fluorophores was added for 15 min (Reagents and Tools Table). That marked the completion of a staining cycle, and antibody stripping was performed at 90 °C in a water bath for 20 min. After the last TSA cycle, DAPI (1 μg/mL in TBS) was added for 10 min, followed by final washes with PBS for 10 min and two times and 15 min wash with TBS-T (Appendix Table S2).

For the TSA using Opal dyes, the Opal 6-colour manual detection kit was followed as per the manufacturer's instructions. The only modifications were that the first antigen retrieval was performed as previously described using a pressure cooker and that a water bath set at 98 °C was used for each stripping cycle.

For the Signal Star, the SignalStar™ Multiplex Immunohisto-chemistry Assay (Cell Signalling Technologies) for Use on BOND RX Fully Automated Research Stainer (Leica Biosystems) was followed as per the manufacturer's instructions (Reagents and Tools Table).

## LCM of PTEN and ARID1A-deficient patches

LCM was performed as described in (Olpe et al, 2021) except for the modifications below. Fresh sections were stained using IHC for the mark of interest, PTEN or ARID1A, the slides were scanned on Leica Aperio AT2, and the location of the patch was mapped on the scanned slide. LCM was performed on a serial section of the stained slide after dewaxing using Leica Multistainer ST5020. WT crypts were also captured to increase the input of DNA. The ratio of deficient crypts to total crypts captured was always 1:2.5, and the minimum patch size captured was a total of ten crypts. Captured crypts were lysed in a volume of 12 μl of Proteinase K buffer from the Arcturus PicoPure DNA kit.

## Fluidigm assay design

The gene of interest, PTEN or ARID1A was imported in the Fluidigm D3 Assay Design and dual coverage primers were designed. There were 45 amplicons used to cover PTEN and 114 to cover ARID1A and they were each pooled in 8 different multiplexing primer pools (Appendix Tables S3, 4).

## Library prep using LP 8.8.6 chip and juno system

Due to the high genomic DNA to mastermix ratio in the LP 8.8.6 chip and the non-purified nature of the DNA lysed using the PicoPure method, an additional Ampure XP bead cleanup (Beckman Coulter™) was required prior to the first PCR. The bead-to-DNA ratio used was 1.8x to clean up the genomic DNA. Then, the library prep was performed using the LP 8.8.6 integrated fluidic circuit (IFC) according to the manufacturer's instructions (Advanta NGS Library Preparation with Targeted DNA Seq Library Assays on the LP 8.8.6 IFC protocol). The only modification was the use of 17 cycles during the first PCR to accommodate for the low DNA input. Samples were submitted for an Illumina NovaSeq run for 150 bp paired-end reads.

## Mutation calling in deficient patches

The SNV mutation calling was performed as mentioned below. Paired-end assembler for Illumina sequences, PANDAseq was used to merge corresponding forward and reverse reads into an in silico amplicon (Masella et al, 2012). The in silico recreated amplicons were then filtered to retain those that began and ended with the expected primer sequence and were the correct overall length. A Perl script was then used to count the number of reads for individual nucleotides (A, C, G or T) at each position in each amplicon. Samples with <500 reads were excluded from downstream analysis. A minimum of 1% VAF was set as the lower detection threshold. The mean variant allele frequency (VAF) and standard deviation was calculated for each position for all nucleotides in all amplicons. The single nucleotide variant (SNV) mutations were then called if the VAF of a particular sample was 3.29 standard deviations above the mean VAF in that specific position (p value <0.001) for each amplicon independently. Only calls that were made in both replicate amplicons were considered. Only calls with two or more variant reads were included. In addition, the presence of insertion and deletions (indels) was also investigated using an alignment-based method, Amplicon-seq (Reagents and Tools Table). The analysis was applied on mutation calls >2% VAF. Mutations that were called in three or more independent samples were excluded.

## Inference of clone dynamics

Clone and crypt counting using DeCryptICS was performed as previously described with sample blinding (Nicholson et al, 2018). The stem cell dynamics and fission rate associated were mathematically modelled as previously described (Nicholson et al, 2018) using the RHClones package (Reagents and Tools Table). A minor modification to the hierarchical model for fission rates was made by instead drawing patient-specific fission rates from a Student's t-distribution according to

$$\rho_i \sim Student T(\nu, \mu, \sigma).$$

Here $\rho_i$ is the fission rate for patient $i$ and the priors for population parameters $\rho$, $\sigma$ and $\nu$ are

$$\rho \sim Normal(0, 10^{-2})$$

$$\sigma \sim Normal\left(0, 10^{-2}\right)$$

$$\nu \sim Gamma\left(2, 10^{-1}\right).$$

Using a Student's t-distribution in place of a Normal distribution allows for control of heavier tails in the hierarchical model for patient-to-patient variability through the parameter $\nu$. Inferred values for $\nu$ were spread across a wide range suggesting that fission rate priors for different genes can display a substantial departure from the standard normal distribution, including regimes where the variance is not strictly well-defined.

## Organoid culture

Intestinal mucosal biopsies were collected from sigmoid colon from children under 16 years old undergoing diagnostic colonoscopy at Addenbrookes Cambridge Hospital. This study was conducted with informed patient and/or carer consent as appropriate, and with full ethical approval. Organoids were cultured as previously described (Skoufou-Papoutsaki et al, 2023). The only modification to the culture conditions was the addition of synthetic fusion Wnt (N001, IpA Therapeutics).

## CRISPR editing of organoids

A ribonucleoprotein approach was used for the CRISPR editing of human intestinal organoids as previously described (Skoufou-Papoutsaki et al, 2023) with the addition of an optimised ratio of dead (d) to active (a) Cas9 for each guide RNA (Alt-R® S.p. dCas9 Protein V3, 100 μg from IDTDNA). For PTEN, the optimised ratio was 20:80 a:dCas9, while for the ARID1A guide, the ratio was 50:50 a:dCas9. After electroporation, single organoids were picked using an IVF EZ-Grip pipette and a dissecting microscope, which were single-cell dissociated and placed in individual wells allowing for clonal selection. Lines with pure 0, 50 or 100% editing as assessed by ICE Synthego (Conant et al, 2022) were expanded and subjected to downstream analysis.

## Automated Wes analysis

Protein extraction and automated Wes (Biotechne) analysis was performed as previously described (Skoufou-Papoutsaki et al, 2023).

## Organoid wholemounts

The organoid wholemount staining protocol was used as previously described (Skoufou-Papoutsaki et al, 2023).

## RNA extraction and bulk RNA-seq

RNA extraction and RNA-seq library prep was performed as previously described (Skoufou-Papoutsaki et al, 2023).

For the data analysis, gene set enrichment analysis (GSEA) was performed using Msigdbr on all the detected genes ranked based on their log fold change and standard error. Enriched pathways with an FDR <1% were considered. Revigo (Supek et al, 2011) was used to identify redundancy in Gene Ontology Biological Processes (GOBP) gene sets and plot their relationship

and interactions. Cluster representatives (terms remaining after redundancy reduction) were plotted in two-dimensional space by applying multidimensional scaling to a matrix of the GO term semantic similarities. Word clouds were created based on the Revigo scatterplot and simplification of the GO terms that most accurately describe the cluster in a biologically meaningful way. When necessary, synonym clusters that describe similar biological processes were merged for simplicity (e.g. substrate adhesion and cell adhesion merged to create an adhesion cluster). STRING (Szklarczyk et al, 2015) was used to identify interaction networks for the DEGs in ARID1A heterozygous vs WT organoids.

## Image analysis

Image analysis was performed on Quantitative Pathology & Bioimage Analysis (QuPath) and Indica Labs Halo Image Analysis Platform for the FFPE tissue and Fiji for the organoid wholemounts. The fluorescence intensity of staining-deficient crypts and five WT neighbouring crypts was calculated on QuPath. For proliferation/differentiation markers, positive cells were manually counted in staining-deficient crypts and five WT neighbouring crypts in MCM2 + CA2- areas (crypt bottom).

For the immune cell analysis on Halo, a random forest crypt-stroma classifier was trained, and a cell segmentation algorithm (Module: Indica Labs HighPlex FL v4.1.3) was used to count the number of positive cells in either the crypt or stromal area. Only ARID1A patches of >5 deficient crypts were included in the analysis. For the Opal analysis, cells were counted within the patch or in a radius of 300 μm around the WT patch. For the Signal Star analysis, cells were counted within the patch and an adjacent (>300-μm away) randomly selected WT area of the same size. Counts were normalised for the crypt or stromal area.

For the organoid wholemount analysis, the mean fluorescence intensity of each organoid was measured using the measure function in Fiji. The background signal and bleaching from the DAPI channel was subtracted by measuring the fluorescence intensity in areas with no signal or areas with DAPI-only signal.

## Estimated mutational burdens and percentage of tumours explained by the gene being a passenger

Inferred stem cell dynamics and fission rates population parameters were used to calculate the estimated mutational burden in normal colon $B(t)$ at a given age $t$ according to the formula

$$B(t) = \sum_{n=1}^{\infty} nF_n(t).$$

Here $F_n(t)$ is the distribution for the probability of a patch of mutated crypts of size $n$ at age $t$ calculated from population estimates for the slope of monoclonal accumulation $\triangle C_{fix}$ and fission rate $\rho$ for that marker given by

$$F_n(t) = \triangle C_{fix} \frac{\left(1 - e^{-\rho t}\right)^n}{\rho n}.$$

In practice, the infinite sum for $B(t)$ was replaced by a finite sum over an extremely large number (in this case 500,000). 95% credible

intervals (CIs) for $B(t)$ were calculated using the same formula replaced with 95% CIs on population parameter value estimates $\triangle C_{fix}$ and $\rho$ for each marker.

To compare estimated mutational burden in normal epithelial with observed mutational burden in CRC, the proportion of colorectal tumours (primary samples not cultured) containing a truncating mutation (i.e. substitution-nonsense, frameshift, deletion-frameshift, insertion-frameshift, whole gene deletion) were extracted from the COSMIC database for each marker gene. The percentage of tumours explained by the gene being a passenger was then calculated by dividing the value for the simulated normal mutational burden at age 70 by the CRC frequency of that gene. Error bars displayed in the corresponding plot are calculated using the 95% credible intervals of the simulated mutational burden.

## Data availability

The bulk RNA-seq data generated in this study are publicly available at Gene Expression Omnibus (GEO) with accession number GSE266990. https://www.ncbi.nlm.nih.gov/geo/query/acc.cgi?acc=GSE266990.

The source data of this paper are collected in the following database record: biostudies:S-SCDT-10_1038-S44319-025-00373-0.

## Peer review information

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

## Acknowledgements

We would like to thank the Histopathology, Genomics, Light Microscopy and Bioinformatics Core at the CRUK Cambridge Institute for their help with the project. This study was funded by the Wellcome Trust (103805, 102160/Z/13/Z) and Cancer Research UK Cambridge Institute, University of Cambridge (A24456) and we would like to thank them for making the work possible.

## Author contributions

**Nefeli Skoufou-Papoutsaki**: Conceptualisation; Data curation; Formal analysis; Funding acquisition; Validation; Investigation; Visualisation; Methodology; Writing—original draft; Writing—review and editing. **Sam Adler**: Investigation; Methodology. **Shenay Mehmed**: Validation; Investigation. **Claire Tume**: Investigation; Methodology. **Cora Olpe**: Methodology. **Edward Morrissey**: Software. **Richard Kemp**: Formal analysis. **Anne-Claire Girard**: Project administration. **Elisa B Moutin**: Writing—review and editing. **Chandra Sekhar Reddy Chilamakuri**: Formal analysis. **Jodi L Miller**: Investigation. **Cecilia Lindskog**: Resources. **Fabian Werle**: Investigation. **Kate Marks**: Resources. **Francesca Perrone**: Resources. **Matthias Zilbauer**: Resources. **David S Tourigny**: Formal analysis; Writing—review and editing. **Douglas J Winton**: Supervision; Funding acquisition; Writing—original draft; Writing—review and editing.

Source data underlying figure panels in this paper may have individual authorship assigned. Where available, figure panel/source data authorship is listed in the following database record: biostudies:S-SCDT-10_1038-S44319-025-00373-0.

## Disclosure and competing interests statement

The authors declare no competing interests.

# Expanded View Figures

**Figure EV1.   Antibody screen for establishment of tumour suppressor gene as a clonal mark.**

(**A**) Failed target- SOX9, not exhibiting uniform expression across the crypt axis. (**B**) Failed target- TCF7L2, identified clone not detected with independent antibody. (**C**) Failed target- FBXW7, antibodies does not separate het and KO samples. VillinCreERT;FBXW7 mouse model used. Human HEK293T cells where FBXW7 was knocked out using CRISPR/Cas9. (**D**) Genetic KO validation of antibodies used for PTEN, SMAD4 and ARID1A. VillinCreERT promoter used for mouse models. (**E**) Independent antibody validation of detected PTEN, SMAD4 and ARID1A clones. Dashed borders within images indicate PTEN, SMAD4 or ARID1A-deficient clones. Data information: All scale bars indicate 50 μm.

▶

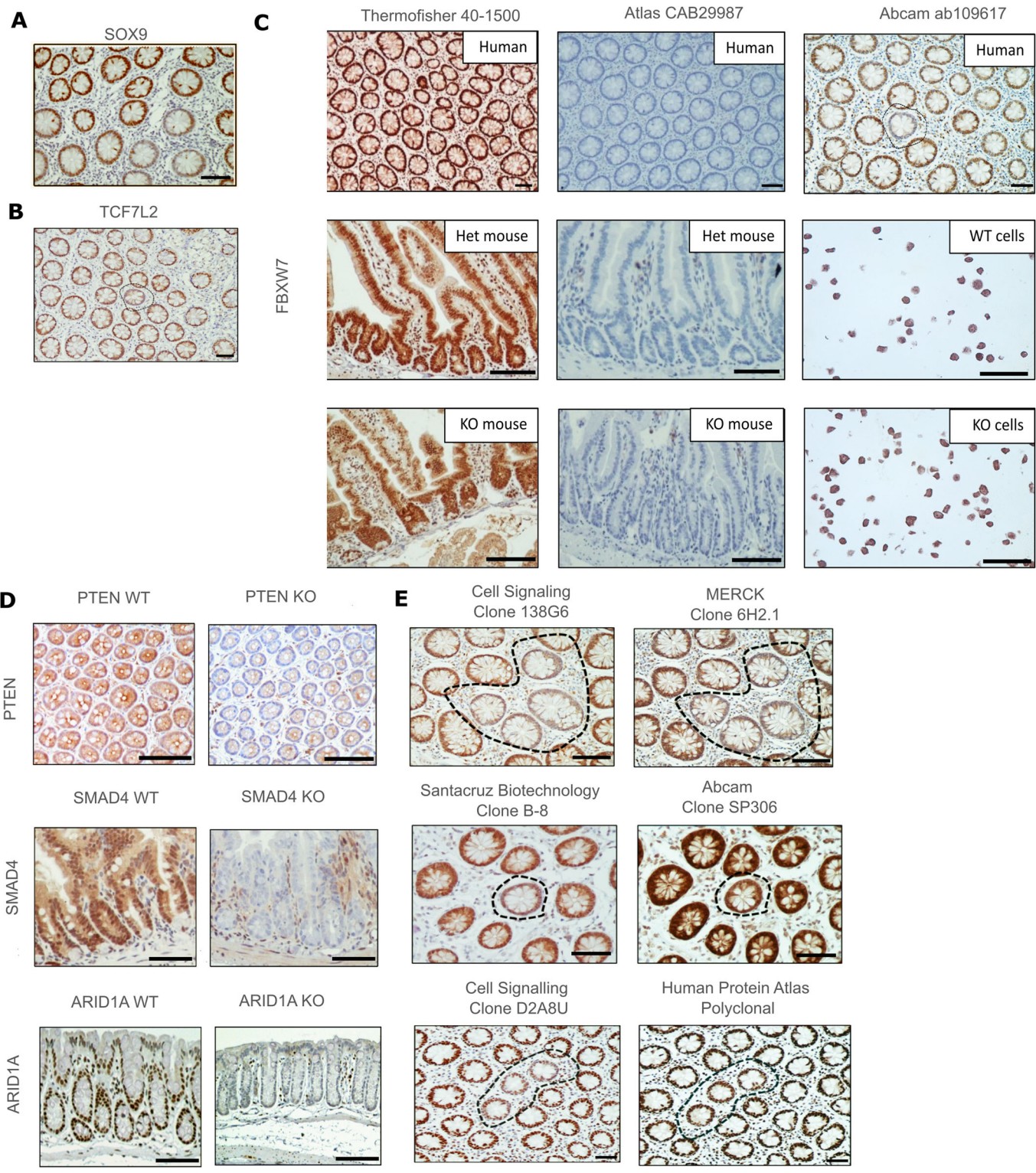

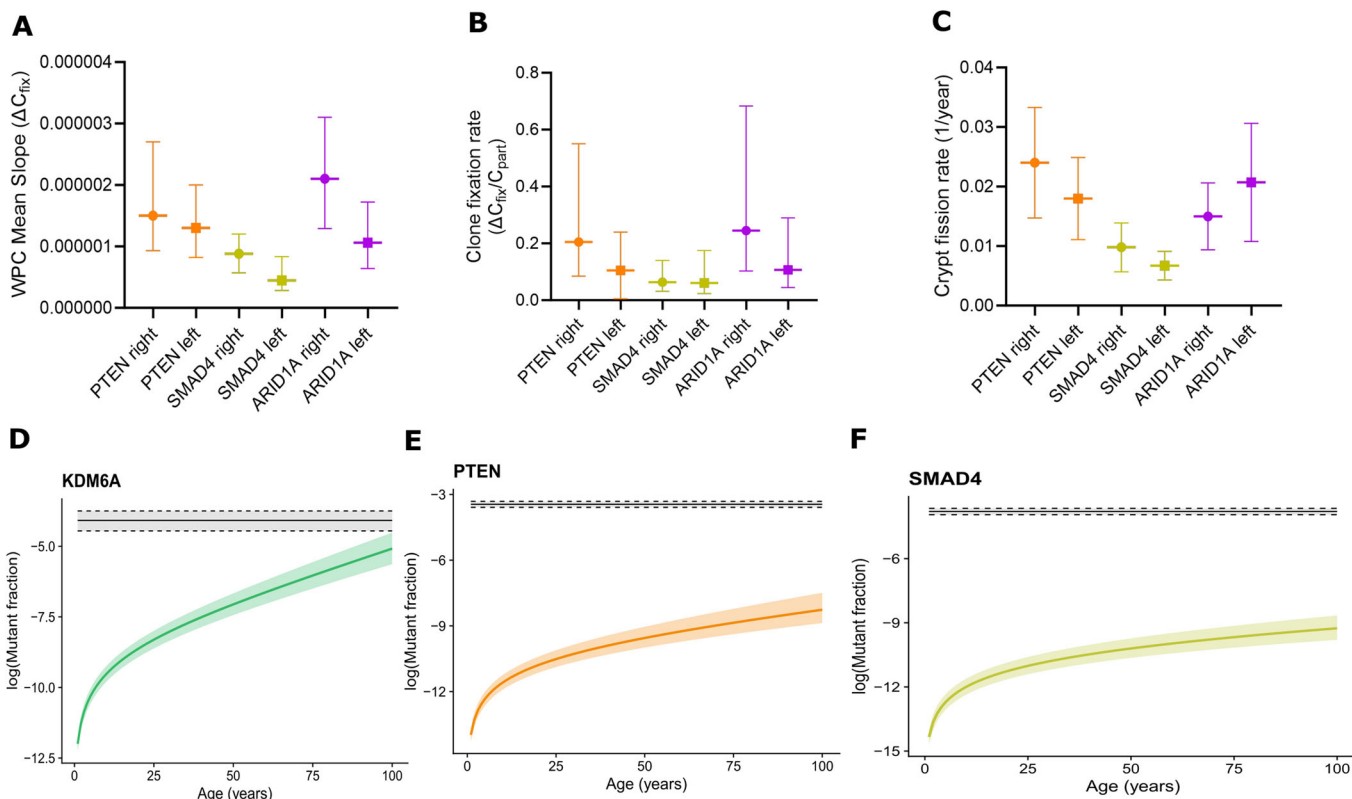

**Figure EV2. Clone dynamics and mutational burden.**

(A–C) Left vs Right side of colon differences. (A) Slope (ΔCfix). (B) Clone fixation rate (ΔCfix/Cpart). (C) Fission rate. (D, E) Predicted mutational burden in normal tissue based on clone dynamics compared to the frequency of truncating mutations for that gene in colorectal cancer in COSMIC. (D) KDM6A, (E) PTEN, (F) SMAD4. Data information: PTEN: $N = 103$ patients and 2,492,712 total crypts, SMAD4: $N = 88$ patients and 1,661,059 crypts, ARID1A: $N = 101$ patients, $N = 1,990,179$ crypts. Data presented as mean. Coloured shading indicates 95% credible intervals. Dotted black lines indicate 95% confidence intervals.

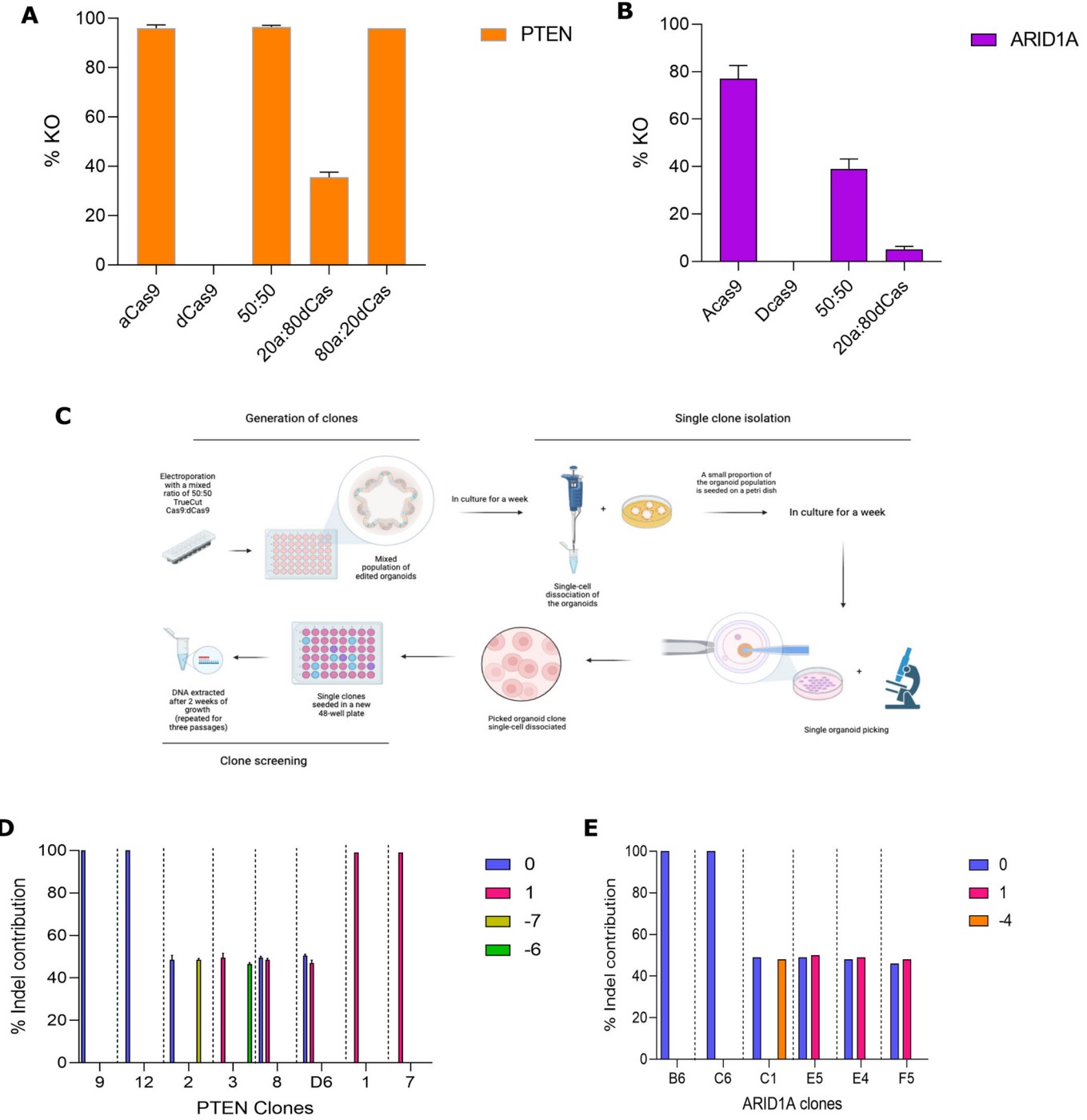

**Figure EV3.  CRISPR-Cas9 ribonucleoprotein-based editing to generate organoids with heterozygous mutations.**

(**A, B**) The ratio of active Cas9 (aCas9) to dead Cas9 (dCas9) was trialled for PTEN and ARID1A guides. (**A**) 20a:80dCas ratio was used for PTEN. (**B**) A 50:50 ratio was used for the ARID1A guide. (**A, B**) $N = 3$ biological and 2 technical replicates. (**C**) Experimental outline of generation of heterozygous organoids and single clone picking. (**D**) Indel contribution for generated PTEN clones. 0 indicates the WT allele. (**E**) Indel contribution for ARID1A clones. 0 indicates the WT allele. (**D, E**) The bar shows individual clones sequenced in forward and reverse directions (error bars). Data information: Data presented as mean ± SD.

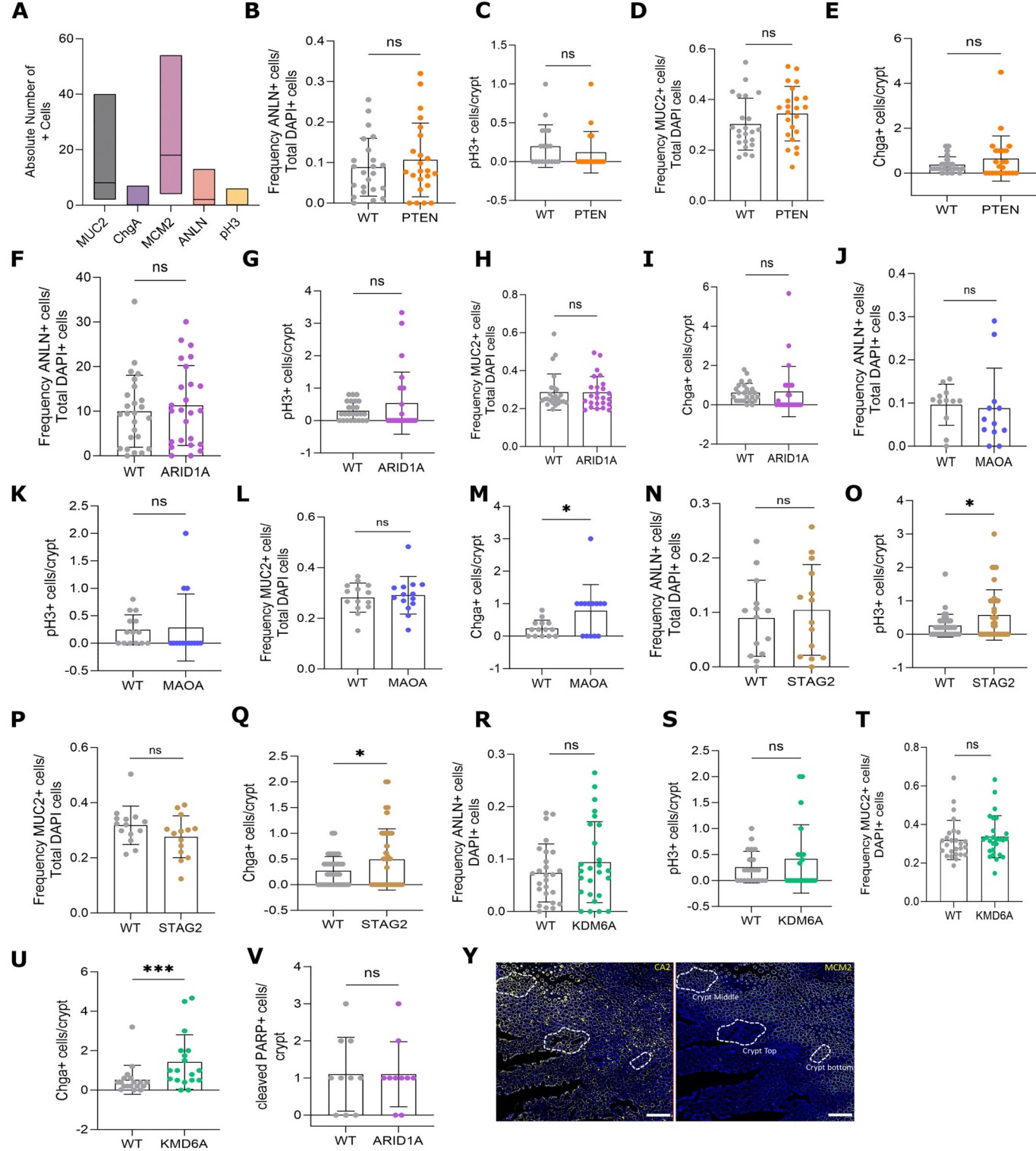

◄ **Figure EV4. Quantification of proliferation and differentiation markers in deficient clones.**

(A) Number of positive cells per crypt for markers used in the panel. Applied on WT crypts. Box plot shows min, median and max. (B–U) ANLN, S/G phase marker. pH3, mitosis marker. MUC2, goblet cells marker. Chga, enteroendocrine marker. Cleaved PARP1, apoptosis marker. ANLN, ph3 and cleaved PARP1 expressed as a number of positive cells per crypt. MUC2 and ANLN expressed as frequency out of total DAPI cells. (Y) Definition of crypt axis levels based on CA2 and MCM2 staining in *en face* embedded tissue. Different levels within a section are seen due to tissue not being completely flat during processing. CA2 + MCM2- crypt top. CA2 + MCM2+ crypt middle. CA2− MCM2+ crypt bottom. Dashed borders indicate crypt top, middle or top according to marker expression. Data information: Data presented as mean ± SD. All scale bars indicate 400 μm. A paired *t*-test or Wilcoxon test was performed to assess statistical significance based on the type of distribution. (B) $p = 0.297$, (C) $p = 0.203$, (D) $p = 0.175$, (E) $p = 0.355$, (F) $p = 0.178$, (G) $p = 0.597$, (H) $p = 0.56$, (I) $p = 0.406$, (J) $p = 0.32$, (K) $p > 0.999$, (L) $p = 0.541$, (M) $p = 0.017$, (N) $p = 0.36$, (O) $p = 0.025$, (P) $p = 0.075$, (Q) $p = 0.035$, (R) $p = 0.12$, (S) $p = 0.508$, (T) $p = 0.247$, (U) $p = 0.0004$, (V) $p > 0.999$. *$p < 0.05$, ***$p < 0.001$. (B–E) $N = 23$ PTEN and WT clones. (F–I) $N = 25$ ARID1A and WT clones. (J–M) $N = 12$ MAOA and WT clones. (N–Q) $N = 14$ STAG2 and WT clones. (R–U) $N = 26$ KDM6A and WT clones. (V) $N = 10$ ARID1A and WT clones.

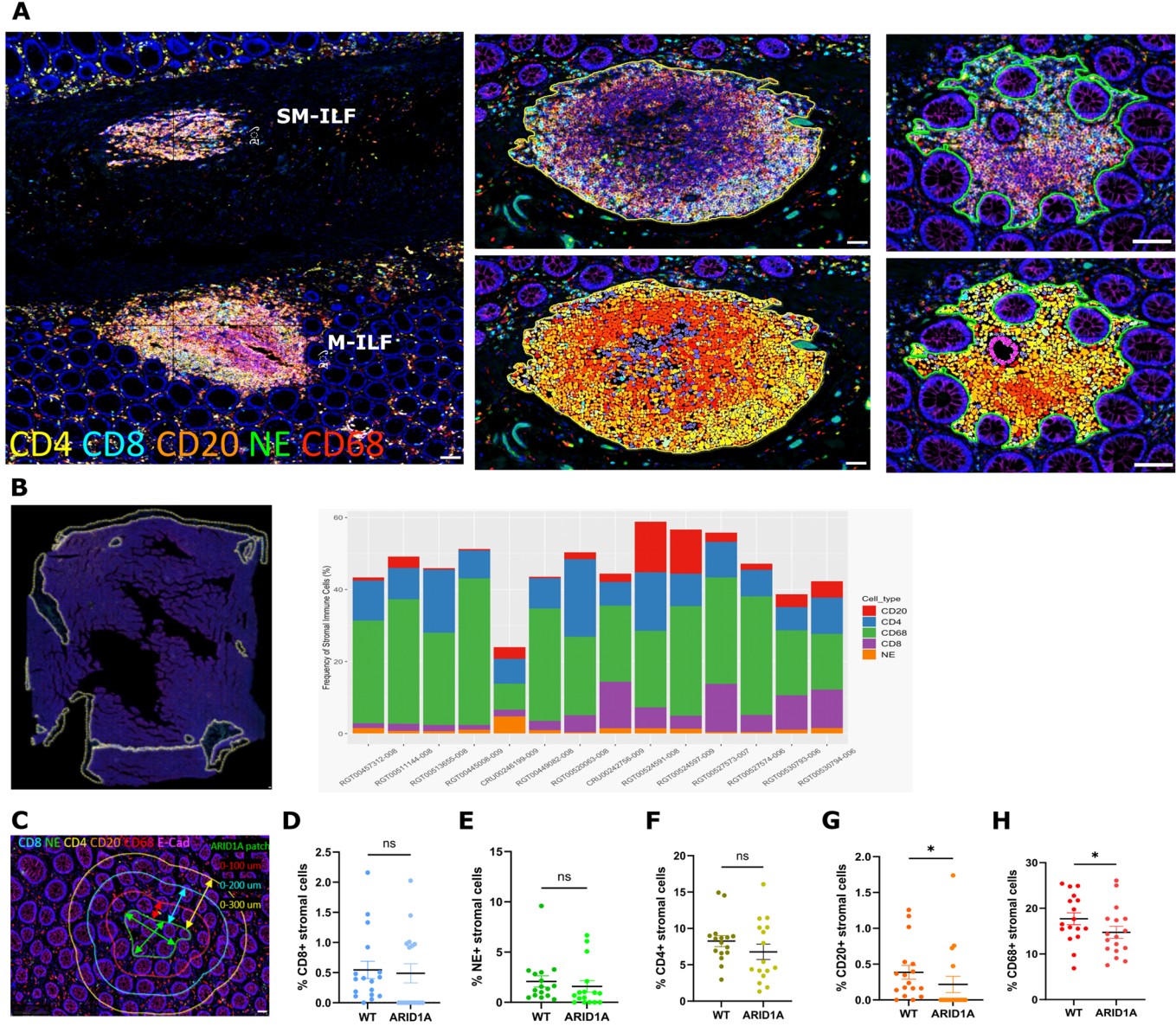

**Figure EV5. Characterisation of immune cell populations in the normal human colon.**

(A) Examples of immune lymphoid follicles in the submucosa (SM-ILF) and the colonic mucosa (M-ILF). The bottom pictures show cell segmentation. (B) Whole-section analysis (excluding immune follicles) showing the contribution of different immune cell types. (C) Infiltration analysis within and around ARID1A patches. (D–H) Percentage of positive stromal cells as defined by trained crypt-stroma random forest classifier (Halo, Indica labs). Cells were measured using the cell segmentation algorithm within the ARID1A patch (>5 crypts) or 300-um radius around the patch (WT). $N = 15$ patients. (D) CD8+ cytotoxic T-cells. (E) NE+ neutrophils. (F) CD4 + T-helper cells. (G) CD20 + B-cells. (H) CD68+ macrophages. Data information: Data presented as mean ± SD. All scale bars indicate 50 μm except for (B) where it indicates 400 μm. A paired Wilcoxon test was performed to assess statistical significance. (D) $p = 0.528$, (E) $p = 0.175$, (F) $p = 0.065$, (G) $p = 0.030$, (H) $p = 0.027$. *$p < 0.05$.

