## [Peer Review File · EMBO Reports]

Haploinsufficient phenotypes promote selection of PTEN and ARID1A deficient clones in human colon

Nefeli Skoufou-Papoutsaki, Sam Adler, Shenay Mehmed, Claire Tume, Cora Olpe, Edward Morrissey, Richard Kemp, Anne-Claire Girard, Elisa Moutin, Chandra Chilamakuri, Jodi Miller, Cecilia Lindskog, Fabian Werle, Kate Marks, Francesca Perrone, Matthias Zilbauer, David Tourigny, and Doug Winton

Corresponding author(s): Doug Winton (doug.winton@cruk.cam.ac.uk)

Review Timeline:

Submission Date:	22nd Oct 24
Editorial Decision:	21st Nov 24
Revision Received:	15th Dec 24
Editorial Decision:	19th Dec 24
Revision Received:	8th Jan 25
Accepted:	10th Jan 25

Editor: Achim Breiling

Transaction Report: Please note that the manuscript was transferred from another journal where it was originally reviewed. Since the original reviews are not subject to EMBO's transparent review process policy, they cannot be published.

Dear Prof. Winton,

Thank you for the submission of your revised manuscript to EMBO reports, previously peer-reviewed at a venue outside EMBO press. I have now received the report from the arbitrator that was asked to evaluate your revised study, which can be found at the end of this email.

As you will see, the arbitrator thinks that these findings are of high interest, that the concerns of the referees have been adequately addressed and that the study should be published in EMBO reports.

Given this feedback, and after going through your p-b-p-response myself, I have decided to proceed with the manuscript. The manuscript needs now to be formatted according to our journal style. Please carefully review the instructions that follow below to prepare your final revised manuscript.

- 1) a .docx formatted version of the final manuscript text (including legends for main figures, EV figures and tables), but without the figures included. Figure legends should be compiled at the end of the manuscript text.
- 2) individual production quality figure files as .eps, .tif, .jpg (one file per figure), of main figures and EV figures. Please upload these as separate, individual files upon re-submission.

- 3) a complete author checklist, which you can download from our author guidelines (<https://www.embopress.org/page/journal/14693178/authorguide>). Please insert page numbers in the checklist to indicate where the requested information can be found in the manuscript. The completed author checklist will also be part of the RPF.

- 4) that primary datasets produced in this study (e.g. RNA-seq, ChIP-seq, structural and array data) are deposited in an appropriate public database. If no primary datasets have been deposited, please also state this in a dedicated section (e.g. 'No primary datasets have been generated and deposited'), see below.

The accession numbers and database should be listed in a formal "Data Availability" section that follows the model below. This is now mandatory (like the COI statement). Please note that the Data Availability Section is restricted to new primary data that are part of this study. This section is mandatory. As indicated above, if no primary datasets have been deposited, please state this in this section

Data availability

5) We now request the publication of original source data with the aim of making primary data more accessible and transparent to the reader. Our source data coordinator will contact you to discuss which figure panels we would need source data for and will also provide you with helpful tips on how to upload and organize the files.

6) Our journal encourages inclusion of *data citations in the reference list* to directly cite datasets that were re-used and obtained from public databases. Data citations in the article text are distinct from normal bibliographical citations and should directly link to the database records from which the data can be accessed. In the main text, data citations are formatted as follows: "Data ref: Smith et al, 2001" or "Data ref: NCBI Sequence Read Archive PRJNA342805, 2017". In the Reference list, data citations must be labeled with "[DATASET]". A data reference must provide the database name, accession number/identifiers and a resolvable link to the landing page from which the data can be accessed at the end of the reference. Further instructions are available at: <http://www.embopress.org/page/journal/14693178/authorguide#referencesformat>

7) Regarding data quantification and statistics, please make sure that the number "n" for how many independent experiments were performed, their nature (biological versus technical replicates), the bars and error bars (e.g. SEM, SD) and the test used to calculate p-values is indicated in the respective figure legends (also for EV and Appendix figures). Please also check that all the p-values are explained in the legend, and that these fit to those shown in the figure. Please provide statistical testing where applicable. Please avoid the phrase 'independent experiment', but clearly state if these were biological or technical replicates. Please also indicate (e.g. with n.s.) if testing was performed, but the differences are not significant. In case n=2, please show the data as separate datapoints without error bars and statistics. See also: <http://www.embopress.org/page/journal/14693178/authorguide#statisticalanalysis>

If n<5, please show single datapoints for diagrams. Please add to each legend (main, EV figures, Appendix, where applicable) a 'Data Information' section explaining the statistics used or providing information regarding replicates and scales. See: <https://www.embopress.org/page/journal/14693178/authorguide#figureformat>

8) Please add scale bars of similar style and thickness to microscopic images, using clearly visible black or white bars (depending on the background). Please place these in the lower right corner of the images themselves. Please do not write on or near the bars in the image but define the size in the respective figure legend.

9) Please also note our reference format:

10) We updated our journal's competing interests policy in January 2022 and request authors to consider both actual and perceived competing interests. Please review the policy <https://www.embopress.org/competing-interests> and update your competing interests if necessary. Please name this section 'Disclosure and Competing Interests Statement' and put it after the Acknowledgements section.

11) We now use CRediT to specify the contributions of each author in the journal submission system. CRediT replaces the author contribution section. Please use the free text box to provide more detailed descriptions and do NOT provide your final manuscript text file with an author contributions section. See also our guide to authors: <https://www.embopress.org/page/journal/14693178/authorguide#authorshippinguidelines>

12) All Materials and Methods need to be described in the main text using our 'Structured Methods' format, which is required for all research articles. According to this format, the Methods section should include a Reagents and Tools Table (listing key reagents, experimental models, software, and relevant equipment and including their sources and relevant identifiers), uploaded as separate file, and a Methods section in which we encourage the authors to describe their methods using a step-by-step protocol format with bullet points, to facilitate the adoption of the methodologies across labs. More information on how to adhere to this format as well as downloadable templates (.doc) for the Reagents and Tools Table can be found in our author guidelines (section 'Structured Methods'):

13) Please add up to 5 keywords to the manuscript and order the sections like this, using these names:
Title page - Abstract - Keywords - Introduction - Results - Discussion - Methods - Data availability section - Acknowledgements (including funding information) - Disclosure and Competing Interests Statement - References - Figure legends - Expanded View Figure legends

14) Please provide the abstract written in present tense throughout.

15) Please make sure that all the funding information is also entered into the online submission system and that it is complete and similar to the one in the acknowledgement section of the manuscript text file.

In addition, I would need from you uploaded separately:

Arbitrator:

After evaluating the revised manuscript and the referees' comments and the point by point reply by the authors, my judgement is that the manuscript is adequately improved.

I can see the hesitation from some of the reviewers regarding the lack of functional or mechanistic insight regarding the mode of clonal expansion of pre-malignant clones. Yet I think that the unique data that is collected here using human samples to infer clonal dynamics in human tissues outweighs this critique. There are a multitude of murine lineage tracing studies but related data in humans remains rare. In addition, the topic that is studied here, goes to the heart of what an oncogenic driver is (oncogene, tumor suppressor gene loss) and I think this is a very elegant discussion of this concept based on actual human data in the normal intestine and colorectal cancer. I recommend publication.

Authors' response to the referee comments from the journal outside EMBO press not shown.

Dear Prof. Winton,

Thank you for the submission of your revised manuscript to our editorial offices. Before I can proceed with formal acceptance, I have these further editorial requests I ask you to address in a final revised manuscript:

- Please up to 5 keywords to the manuscript and order the sections like this, using these names:

Title page - Abstract - Keywords - Introduction - Results - Discussion - Methods - Data availability section - Acknowledgements (including funding information) - Disclosure and Competing Interests Statement - References - Figure legends - Expanded View Figure legends

- The nomenclature for the EV figures is Figure EVx. Please use this for the filenames and in the legends. Please check again that also the callouts follow this nomenclature.

- Please make sure that all figure panels and tables (main, EV and Appendix figures) are called out separately and sequentially. Presently, there are callouts for Suppl. Methods Table 1-5, but no such tables uploaded and callouts for Appendix Table S2-S4 are missing. Please check. Please also add callouts for the reagents and tools table.

- Please check that the number "n" for how many independent experiments were performed, their nature (biological versus technical replicates), the bars and error bars (e.g. SEM, SD) and the test used to calculate p-values is indicated in the respective figure legends (main, EV and Appendix figures). Please also check that all the p-values are explained in the legend, and that these fit to those shown in the figure. Please provide statistical testing where applicable. Please avoid the phrase 'independent experiment', but clearly state if these were biological or technical replicates. Please also indicate (e.g. with n.s.) if testing was performed, but the differences are not significant. In case n=2, please show the data as separate datapoints without error bars and statistics. See also:

<http://www.embopress.org/page/journal/14693178/authorguide#statisticalanalysis>

If $n < 5$, please show single datapoints for diagrams. Could testing be also done for the diagrams in panels 2D, 2F, 2G, 2J, 3B, 3M, 3N and EV2A-C? Moreover:

- Please note that the exact p values are not provided in the legend of figure 4P.

- Please note that the box plots need to be defined in terms of minima, maxima, centre, - Please note that information related to n is missing in the legends of figures 2D, F, G, J; 3B, I, J, M, N; EV2 A-C; EV3 A, B, D; EV4 B-V.

- Please note that the error bars are not defined in the legends of figures 4K, L, M, N, O; 5D, E, F, G, H, I, K, O; EV3 A, B, D; EV4 B-V; 5D-H.

- Please note that the measure of center for the error bars needs to be defined in the legends of figures 2D, F, G, J; 3B, I, J, M, N; EV2 A-C.

- Please note that the dashed borders are not defined in the legend of figures 2A-C, E; 4J, EV1 E, EV4 Y. This needs to be rectified.

- Please note that the white borders are not defined in the legend of figures 3C, 4I. This needs to be rectified."

- Please add to each legend (main, EV figures, Appendix figures where applicable) a 'Data Information' section explaining the statistics used or providing information regarding replicates and scales. See:

- Please add scale bars of similar style and thickness to all microscopic images (main, EV and Appendix figures), using clearly visible black or white bars (depending on the background). Please place these in the lower right corner of the images themselves. Please do not write on or near the bars in the image but define the size in the respective figure legend. Presently, some scale bars are hard to see (e.g. EV4Y) or are missing (EV1, Ev5). Please check.

- Please provide the Appendix as pdf file and add a title to the title page ("Appendix for ...").

- Please add the manuscript number to the general info table of the author checklist.

- Thank you for providing the requested source data. It seems the source data for 3N is missing. The file in the folder 3L-M does not contain a TAB for 3N. Please check. Moreover, please provide a source data checklist without comments directly included in the PDF, but in the comments boxes at the end of the pages.

- Please provide the schematic summary figure in jpeg or tiff format with the exact width of 550 pixels and a height of not more than 400 pixels.

Best,

All editorial and formatting issues were resolved by the authors.

Prof. Doug Winton
Cancer Research UK Cambridge Institute
Li Ka Shing Centre
University of Cambridge
Robinson Way
Cambridge CB2 0RE
United Kingdom

Dear Prof. Winton,

I am very pleased to accept your manuscript for publication in the next available issue of EMBO reports. Thank you for your contribution to our journal.

Yours sincerely,
